# Pareto Variational Autoencoder

**Mincheol Cho**[1*]   **Yedarm Seong**[2*]   **Joong-Ho Won**[1,2]
[1]Department of Statistics, Seoul National University
[2]Interdisciplinary Program in Artificial Intelligence, Seoul National University
`{code1478,mybirth0407,won.j}@snu.ac.kr`

## Abstract

This paper introduces a new class of multivariate power-law distributions—the symmetric Pareto (symPareto) distribution—which can be viewed as an $\ell_1$-norm-based counterpart of the multivariate $t$ distribution, with the motivation of capturing the heavy tail of the target distribution in generative modeling and bringing robustness to noise in downstream tasks such as image denoising. The symPareto distribution possesses many attractive information-geometric properties with respect to the $\gamma$-power divergence that is a natural alternative to the Kullback-Leibler divergence, the core of the conventional variational autoencoder (VAE) models, for power families. Leveraging on the joint minimization view of variational inference, this paper proposes the ParetoVAE, a probabilistic autoencoder that minimizes the $\gamma$-power divergence between two statistical manifolds. ParetoVAE employs the symPareto distribution for both prior and encoder, with flexible decoder options including multivariate $t$ and symPareto distributions. Empirical evidences demonstrate the effectiveness of ParetoVAE across multiple domains through varying the types of the decoder. The $t$ decoder achieves superior performance in sparse, heavy-tailed data reconstruction and word frequency analysis; the symPareto decoder enables robust high-dimensional denoising.

## 1 Introduction

For over a decade since its introduction, the Variational Autoencoder (VAE, Kingma & Welling, 2013) has stood the test of time as a fundamental probabilistic generative model for scalable probabilistic inference and representation learning. Conventional VAEs typically employ exponential-family distributions, most notably multivariate Gaussian, for their probabilistic model due to their mathematical tractability. However, the exponential tail assumption often falls short when modeling real-world data that exhibit heavy tails and extreme events. For example, scale-free network degree distributions (Barabási & Albert, 1999) and long-tailed class-frequency distributions (Zhang et al., 2023) challenge models built on light-tail assumptions. In particular, the VAE literature has pointed out that Gaussian-based VAEs tend to underestimate tail probabilities, over-regularize latent codes, and fail to capture rare but informative events (Lafon et al., 2023; Kim et al., 2024).

To address these limitations, recent work (Takahashi et al., 2018; Abiri & Ohlsson, 2020; Kim et al., 2024) has explored VAEs based on multivariate Student's $t$ ditribution, yet the latter constitutes only one option among many heavy-tailed distribution families. In particular, classical extreme-value theory suggests that univariate Pareto distribution-based modeling is well suited to capturing tail and extreme behavior (Balkema & de Haan, 1974; Pickands, 1975). Motivated by this, we introduce a new multivariate generalization of the Pareto family—the symmetric Pareto (**symPareto**) distribution— and corresponding heavy-tail VAE framework, named **ParetoVAE**. ParetoVAE incorporates symPareto distributions as key components, overcoming the representational limitations of Gaussian VAEs in extreme-value scenarios and inducing sparsity in the latent embedding space through the inherent $\ell_1$ formulation.

The use of symPareto distributions in conventional VAEs raises computational challenges, especially for the evidence lower bound (ELBO) estimation. Specifically, the Kullback-Leibler (KL) divergence between two symPareto distributions lacks a closed-form expression, necessitating numerical integra-

---

*Equal contribution.

Table 1: Comparison of heavy-tailed VAE variants. Each tuple denotes (reconstruction loss, latent regularization), where the components are either $\ell_1$ or squared $\ell_2$ ($\ell_2^2$).

| Latent distribution (Encoder / Prior) | Decoder distribution (Generator) | |
|---|---|---|
| | Student's $t$ | **SymPareto** |
| Student's $t$ | $(\ell_2^2, \ell_2^2)$ (Kim et al., 2024) | $(\ell_2^2, \ell_1)$ |
| **SymPareto** | $(\ell_1, \ell_2^2)$ | $(\ell_1, \ell_1)$ |

tion that becomes computationally prohibitive as the dimension grows, or Monte Carlo methods that cost extra variability. We address this issue by adopting an information-geometric joint minimization framework based on the $\gamma$-power divergence, serving as a tractable alternative to the KL divergence. This reformulation admits closed-form expressions between power-law distributions, thereby enabling efficient optimization for modeling heavy-tailedness.

This work naturally extends the family of heavy-tailed variants of VAEs from Student's $t$ distributions to symPareto. In particular, we generalize the $t^3$VAE (Kim et al., 2024) structure into a symPareto-based formulation. Table 1 provides an overview of the resulting latent/decoder distribution combinations and their corresponding reconstruction loss/regularization structures (either $\ell_1$ or $\ell_2^2$). It shows how our proposal broadens the $t$-based heavy-tailed VAE family to a symPareto-based classes. Notably, incorporating symPareto distributions induces an $\ell_1$ term in the objective, featuring sparsity and robustness in heavy-tailed VAE models.

**Related Work**   Recognizing the limitations of Gaussian distributions in standard VAEs, many researchers have proposed approaches that modify the ELBO by altering the prior, encoder, or decoder. On the prior side, spike-and-slab priors (Tonolini et al., 2020), Dirichlet priors (Joo et al., 2020), and tilted-Gaussian variants (Floto et al., 2023) have been explored as non-Gaussian alternatives. A distinct line of work incorporates heavy-tailed distributions directly into the VAE components. Student's $t$ decoders improve robust density estimation (Takahashi et al., 2018); heavy-tailed radii and conditional angles target multivariate extremes (Lafon et al., 2023); conditional $t$-based decoders enhance class-conditional generation quality (Bouayed et al., 2025); and spherical Cauchy encoders combined with a uniform prior effectively capture directional structure (Sablica & Hornik, 2025).

Beside modifying the model distributions, other VAE approaches focus on replacing the divergence measure itself. For example, Rényi's $\alpha$-divergence (Li & Turner, 2016), skew-geometric Jensen-Shannon divergence (Deasy et al., 2021), and $\beta$-divergence (Akrami et al., 2022) have been used in VAEs for specific purposes such as robust estimation. Beyond alternative divergences, some approaches directly reformulate the ELBO via joint minimization problem, such as InfoVAE (Zhao et al., 2019) with mutual-information regularization and Coupled VAE (Hao & Shafto, 2023) with entropic optimal transport. Notably, the $t^3$VAE (Kim et al., 2024) introduces a joint minimization framework that leverages the $\gamma$-power divergence and replaces the prior, encoder, and decoder distributions with multivariate $t$ distributions, based on information geometric principles (Amari, 2016; Eguchi, 2021).

Other generative models, especially those based on the generative adversarial network (GAN) framework, have also been developed to explicitly capture heavy-tail behaviors by changing the prior distribution or divergence. Some GAN-based models use power-law priors, such as univariate $t$ (Feder et al., 2020) and Pareto (Huster et al., 2021) distributions. Others apply some exotic divergences, such as the Lipschitz-regularized $\alpha$-divergence (Chen et al., 2024). Furthermore, normalizing flows (Laszkiewicz et al., 2022; Hickling & Prangle, 2024) and diffusion models (Pandey et al., 2024; Lian et al., 2025) have garnered attention for their ability to handle rare or extreme data points by moving beyond the Gaussianity assumption.

## 2 THEORETICAL BACKGROUND

### 2.1 VARIATIONAL AUTOENCODER (VAE)

VAE aims to approximate the true data distribution $p_{\text{data}}(x)$ by modeling the marginal likelihood as $p_\theta(x) = \int p_\theta(x|z)p_Z(z)\,dz$, where $z$ represents a latent variable. Due to the intractability of

directly computing the posterior $p_\theta(z|x)$, VAE employs variational inference to approximate this distribution. The VAE framework consists of two primary components, encoder (inference model) $q_\phi(z|x) \in \mathcal{F}$ and decoder (generative model) $p_\theta(x|z) \in \mathcal{G}$, where $\mathcal{F}$ and $\mathcal{G}$ are spaces of density functions parameterized by deep neural networks. The encoder approximates the posterior $p_\theta(z|x)$, while the decoder generates a sample $x$ conditioned on the latent variable $z$.

The primary objective is to minimize the KL divergence $\mathcal{D}_{\mathrm{KL}}(\cdot \| \cdot)$ between the true posterior $p_\theta(z|x)$ and its approximation $q_\phi(z|x)$ (encoder). This is equivalent to maximizing the evidence lower bound (ELBO) of the marginal log-likelihood, expressed as

$$\mathrm{ELBO}_{\theta,\phi}(x) = \mathbb{E}_{z \sim q_\phi(\cdot|x)}[\log p_\theta(x|z)] - \mathcal{D}_{\mathrm{KL}}(q_\phi(z|x) \| p_Z(z)).$$

Here $p_Z(z)$ refers to the prior density, which is commonly assumed to be a standard $m$-variate Gaussian $p_Z(z) \sim \mathcal{N}_m(0, I)$. Typically, both the encoder and decoder are also chosen as multivariate Gaussian

$$q_\phi(z|x) \sim \mathcal{N}_m(\mu_\phi(x), \Sigma_\phi(x)), \quad p_\theta(x|z) \sim \mathcal{N}_n(\mu_\theta(z), \sigma^2 I).$$

with diagonal encoder variance $\Sigma_\phi(x) = \mathrm{diag}(\sigma_\phi^2(x))$ to simplify computation. As a result of the Gaussianity assumption, the objective function of the VAE can be written as the sum of the mean squared error (MSE, $\mathbb{E}_{z \sim q_\phi(\cdot|x)} \left[ \frac{1}{2\sigma^2} \| x - \mu_\theta(z) \|^2 \right]$) and a KL regularization term.

**VAE as a Joint Minimization Problem**   VAE objective can be viewed as a joint minimization problem between two statistical manifolds (Han et al., 2020). The model distribution manifold $\mathcal{M}_{\mathrm{model}}$ and the data distribution manifold $\mathcal{M}_{\mathrm{data}}$ are defined as

$$\mathcal{M}_{\mathrm{model}} = \{p_\theta(x, z) = p_\theta(x|z)p_Z(z) : \theta \in \Theta\}, \ \mathcal{M}_{\mathrm{data}} = \{q_\phi(x, z) = p_{\mathrm{data}}(x)q_\phi(z|x) : \phi \in \Phi\}.$$

The KL divergence between these manifolds can be expressed as follows:

$$\mathcal{D}_{\mathrm{KL}}(q_\phi \| p_\theta) = -\mathbb{E}_{x \sim p_{\mathrm{data}}}[\mathrm{ELBO}_{\theta,\phi}(x)] - \mathcal{H}(p_{\mathrm{data}}),$$

where $\mathcal{H}(p_{\mathrm{data}})$ is the differential entropy of the data distribution. Therefore, maximizing the ELBO can be reformulated as the following joint minimization problem:

$$(p_{\theta^*}, q_{\phi^*}) = \mathrm{argmin}_{p \in \mathcal{M}_{\mathrm{model}}, q \in \mathcal{M}_{\mathrm{data}}} \mathcal{D}_{\mathrm{KL}}(q \| p). \tag{1}$$

This perspective allows us to interpret VAE as minimizeing the KL divergence between $\mathcal{M}_{\mathrm{model}}$ and $\mathcal{M}_{\mathrm{data}}$, which can be solved by the information-geometric $em$-algorithm (Csiszár, 1984; Han et al., 2020). The key point is that, when constructing VAE structures, one can *bypass an explicit ELBO maximization* and instead directly solve a joint minimization problem. Moreover, this view naturally motivates replacing KL divergence with alternative divergences to move beyond Gaussianity.

## 2.2   DISTRIBUTIONS IN POWER-LAW FAMILIES

A random variable $X$ follows a (univariate) power-law distribution if its probability density or mass function $p(x)$ is proportional to $x^{-\alpha}$, $\alpha > 0$ (Newman, 2005). We introduce some multivariate extensions of power-law distributions relevant to heavy-tailed VAE models.

**Multivariate $t$ distribution**   The $n$-variate Student's $t$ distribution with degrees of freedom $\nu > 0$ has its density function as

$$t_n(x|\mu, \sigma^2 I_n, \nu) = \frac{T_{n,\nu}}{\bar{\sigma}} \left( 1 + \frac{1}{\nu} \left\| \frac{x - \mu}{\sigma} \right\|_2^2 \right)^{-\frac{\nu+n}{2}}, \quad T_{n,\nu} = \frac{\Gamma(\frac{\nu+n}{2})}{(\nu\pi)^{\frac{n}{2}}\Gamma(\frac{\nu}{2})}. \tag{2}$$

where $\bar{\sigma} := \prod_{i=1}^n \sigma_i$. Here $\mu \in \mathbb{R}^n$ and $\sigma \in \mathbb{R}_+^n$ denote the location and scale parameters, respectively, and the division $(x - \mu)/\sigma$ is to be understood element-wise, with a slight abuse of notation. It is widely regarded as a heavy-tailed generalization of the multivariate Gaussian, since it converges to $\mathcal{N}_n(\mu, \sigma^2 I_n)$ as $\nu \to \infty$.

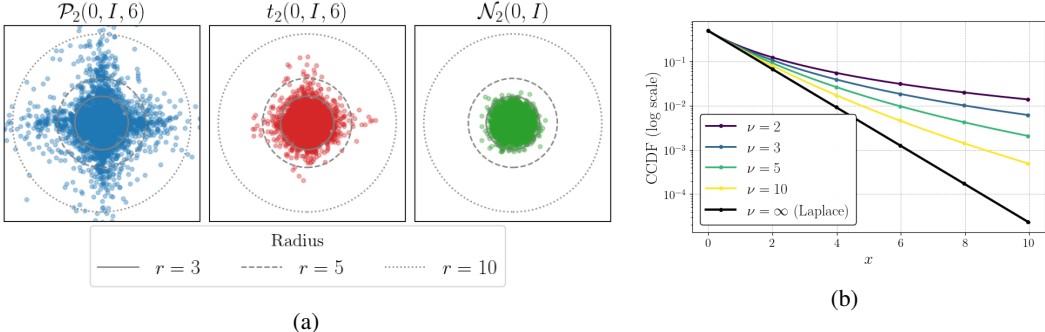

Figure 1: (a) 2D scatter plots of SymPareto (left), Student's $t$ (middle), and Gaussian (right). (b) Log-scale CCDF comparison of SymPareto with varying $\nu$.

**Multivariate Symmetric Pareto distribution**    Although several formulations of the multivariate Pareto distribution have been proposed (Arnold et al., 1993; Hanagal, 1996; Rootzén et al., 2017), most lack explicit density representations suitable for the computation of divergence and are restricted to positive supports. Motivated by Mardia's multivariate Pareto of the first kind (Mardia, 1962), we introduce the multivariate symmetric Pareto distribution (symPareto). Its density is defined as

$$\mathcal{P}_n(x \mid \mu, \sigma, \nu) = \frac{C_{n,\nu,\nu}}{\bar{\sigma}} \left( 1 + \frac{1}{\nu} \left\| \frac{x - \mu}{\sigma} \right\|_1 \right)^{-(\nu+n)}, \quad C_{n,\nu_1,\nu_2} = \frac{\Gamma(\nu_1 + n)}{(2\nu_2)^n \Gamma(\nu_1)}. \tag{3}$$

This distribution can be regarded as a heavy-tailed analogue of the product of univariate Laplace distributions, or as an $\ell_1$-norm–based counterpart of the multivariate $t$ distribution. Further properties of the symPareto distribution are provided in Appendix A.

Figure 1 illustrates characteristic behaviors of the proposed symPareto distribution. Figure 1a displays scatter plots of 5,000 samples drawn from the symPareto, Student's $t$, and Gaussian distributions in two dimensions. Most Gaussian samples concentrate within radius $r \leq 3$ due to the exponentially decaying tail, whereas Student's $t$ produces noticeably more samples beyond this range. The symPareto distribution exhibits even heavier tails, with many extreme samples lying outside radius 5 or 10. Moreover, the samples of symPareto tend to align the coordinates, producing a cross-like shape. This reflects the $\ell_1$-norm structure in its density.

Figure 1b shows the complementary cumulative distribution functions (CCDFs) of the symPareto distribution on a log scale for various values of $\nu$. For small $\nu$, the CCDF exhibits polynomial decay, indicating heavy tails. As $\nu$ increases, the tail becomes lighter, and in the limit $\nu \to \infty$ the symPareto converges to the Laplace distribution, which decays exponentially.

### 2.3 Information Geometry and $\gamma$-power Divergence

In this section, we briefly introduce the $\gamma$-power divergence as an alternative to the KL divergence from an information-geometric perspective. More formal and detailed exposition of these concepts can be found in Amari (2016).

**Divergence in information geometry**    In information geometry, a family of probability distributions is regarded as a statistical manifold $\mathcal{S}$, and a specific distribution in the family is a point on $\mathcal{S}$. To quantify the discrepancy between two points on $\mathcal{S}$, one typically employs a differentiable function $\mathcal{D} : \mathcal{S} \times \mathcal{S} \to [0, \infty)$, called a divergence. A divergence not only measures separation of points but also induces geometric structures on $\mathcal{S}$, including an affine connection $\nabla$ that governs how tangent vectors are transported between nearby points. The connection $\nabla$ in turn defines $\nabla$-geodesics, i.e., one-dimensional submanifolds that generalize the notion of straight lines on the statistical manifold.

Formally, each $\nabla$-geodesic can be expressed linearly with respect to affine coordinate systems associated with $\nabla$. In this sense, the manifold $\mathcal{S}$ is said to be $\nabla$-*flat*. This property enables powerful tools of geometric analysis, such as joint minimization algorithms (Csiszár, 1984; Han et al., 2020).

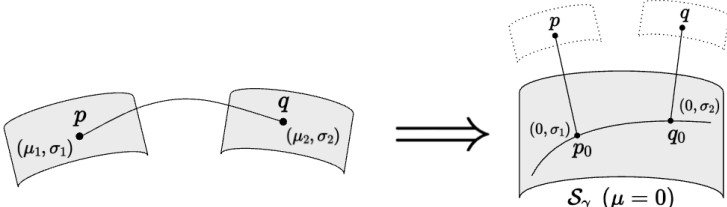

Figure 2: Illustration of two symPareto manifolds with different $\mu$ and their translation to the $\gamma$-flat manifold $S_\gamma$ for analyzing the $\gamma$-power divergence.

**KL divergence**  A well-known example of a divergence is the KL divergence. One of the corresponding geodesics is the $e$-geodesic ($e$ for exponential), defined by

$$l_e(x) \propto \exp((1-t)\log p(x) + t \log q(x)), \quad t \in [0,1].$$

In particular, $e$-geodesics characterize distributions in the exponential families, implying that these families form $e$-flat submanifolds. Thus, the KL divergence naturally induces geometric structure with respect to exponential families. However, not all statistical manifolds exhibit such family. For instance, for distributions in the power-law family, such as the $t$ and generalized Pareto distributions, the KL divergence does not induce a dually flat geometric structure.

**$\gamma$-power divergence**  A divergence suitable for the power-law families is the $\gamma$-power divergence (Kim et al., 2024), a generalization of the notion due to Eguchi (2021). It is defined as

$$\mathcal{D}_\gamma(q \,\|\, p) := \gamma^{-1}\mathcal{C}_\gamma(q,p) - \gamma^{-1}\mathcal{H}_\gamma(q), \quad \gamma \in (-1,\infty) \setminus \{0\}$$

$$\mathcal{H}_\gamma(p) := -\|p\|_{1+\gamma} = -\left(\int p(x)^{1+\gamma}dx\right)^{\frac{1}{1+\gamma}}, \quad \mathcal{C}_\gamma(q,p) := -\int q(x)\left(\frac{p(x)}{\|p\|_{1+\gamma}}\right)^\gamma dx.$$

where $\mathcal{H}_\gamma(\cdot)$ and $\mathcal{C}_\gamma(\cdot,\cdot)$ are called the $\gamma$-power entropy and $\gamma$-power cross-entropy, respectively. This divergence parallels the $e$-geodesic for the KL divergence by introducing the $\gamma$-power geodesic, which induces $\gamma$-flat submanifolds as

$$\mathcal{S}_\gamma = \left\{ p_\theta(x) \propto (1 + \gamma\theta^\top s(x))^{\frac{1}{\gamma}} : \theta \in \Theta \right\}, \tag{4}$$

where $s(x)$ is the sufficient statistic of the given distribution. When the symPareto has $\mu = 0$, $s(x) = |x| \in \mathbb{R}^n$ is a valid sufficient statistic. In this case, the symPareto density can be expressed in the $\gamma$-flat form (4) with $\gamma = -\frac{1}{\nu+n}$.

**Upper bound of $\gamma$-power divergence for flatness**  However, when $\mu \neq 0$, the symPareto distribution no longer admits an sufficient statistic $s(x)$ of the required form even though the $\gamma$-power divergence between two symPareto distributions with different $\mu$ can be written in closed form. Consequently, $\gamma$-flatness is not preserved in the noncentral symPareto distributions.

To handle this case, we consider a translation of the distributions. By shifting the location parameters $\mu_1, \mu_2$ to the origin, the corresponding zero-centered distributions $p_0 = \mathcal{P}_n(0, \sigma_1, \nu)$ and $q_0 = \mathcal{P}_n(0, \sigma_2, \nu)$ lie on the $\gamma$-flat manifold $S_\gamma$ with $\gamma = -\frac{1}{\nu+n}$. Figure 2 illustrates two manifolds of the symPareto distribution with different $\mu$, and their translation to the $\gamma$-flat manifold $S_\gamma$. We then obtain an upper bound of $\mathcal{D}_\gamma(p \,\|\, q)$ by $\gamma$-power divergence between $p_0$ and $q_0$ together with an additional term reflecting the translation cost in terms of the $\ell_1$-distance between $\mu_1$ and $\mu_2$ relative to the scale.

**Theorem 2.1** (Upper bound for $\gamma$-power divergence between two noncentral symPareto distributions)**.** *Let $n \geq 1$, $\nu > 1$, and $\gamma = -\frac{1}{\nu+n}$. For $p_0 = \mathcal{P}_n(0, \sigma_1, \nu)$ and $q_0 = \mathcal{P}_n(0, \sigma_2, \nu)$, the $\gamma$-power divergence between $p_0$ and $q_0$ is finite when $\nu > 1$ and can be expressed in closed-form*

$$\mathcal{D}_\gamma(p_0 \,\|\, q_0) = C'\left[\bar{\sigma}_2^{-\frac{\gamma}{1+\gamma}}\left(1 + \frac{1}{\nu-1}\left\|\frac{\sigma_1}{\sigma_2}\right\|_1\right) - \bar{\sigma}_1^{-\frac{\gamma}{1+\gamma}}\left(\frac{\nu+n-1}{\nu-1}\right)\right], \tag{5}$$

where $C' := C'(n, \nu, \gamma) = (\nu + n)C_{n,\nu-1,\nu}^{\frac{\gamma}{1+\gamma}}$. *Furthermore, for two noncentral symPareto distributions* $p = \mathcal{P}_n(\mu_1, \sigma_1, \nu)$ *and* $q = \mathcal{P}_n(\mu_2, \sigma_2, \nu)$, *the divergence* $\mathcal{D}_\gamma(p \,\|\, q)$ *has an upper bound*

$$\mathcal{D}_\gamma(p \,\|\, q) \leq \mathcal{D}_\gamma(p_0 \,\|\, q_0) + \beta \left\| \frac{\mu_1 - \mu_2}{\sigma_2} \right\|_1, \tag{6}$$

*with* $\beta = \left(1 + \frac{n}{\nu}\right) C_{n,\nu-1,\nu}^{\frac{\gamma}{1+\gamma}} \bar{\sigma}_2^{-\frac{\gamma}{1+\gamma}}$. *In* (6), *equality holds when* $\mu_1 = \mu_2$.

# 3 THE PARETO VARIATIONAL AUTOENCODER

## 3.1 THE PARETOVAE STRUCTURE

In this section, we introduce the ParetoVAE framework and compute the corresponding $\gamma$-loss objective via the $\gamma$-power divergence. All detailed derivations and proofs are in Appendix B.

**Prior, decoder, and encoder**  Motivated by (1), we set a minimization problem for VAE as minimizing the $\gamma$-power divergence between two joint manifolds:

$$(q_{\phi^*}, \, p_{\theta^*}) = \mathrm{argmin}_{\theta \in \Theta, \, \phi \in \Phi} \mathcal{D}_\gamma(q_\phi \,\|\, p_\theta), \tag{7}$$

where $q_\phi(x, z) = q_\phi(z|x)p_{\mathrm{data}}(x)$ and $p_\theta(x, z) = p_\theta(x|z)p(z)$. Under this problem setup, the construction of ParetoVAE starts from the heavy-tailed joint decoder model:

$$p_\theta(x, z) \propto \left[ 1 + \frac{1}{\nu} \left( \|z\|_1 + \frac{\|x - \mu_\theta(z)\|_2^2}{\sigma^2} \right) \right]^{-\frac{2(\nu+m)+n}{2}}. \tag{8}$$

From (8), we deduce the prior $p(z)$ and decoder $p_\theta(x \mid z)$:

$$p(z) = \mathcal{P}_m(z \mid 0, \mathbf{1}_m, \nu), \quad p_\theta(x \mid z) = t_n \left( x \mid \mu_\theta(z), \frac{\nu + \|z\|_1}{2(\nu + m)}\sigma^2 I_n, \nu + m \right)$$

For the encoder, we also adopt a symPareto distribution consistent with the prior, while increasing the degree of freedom by $n$ to reflect the contribution of the data dimension:

$$q_\phi(z|x) = \mathcal{P}_m(z \mid \mu_\phi(x), \sigma_\phi(x), \nu + n/2).$$

**Reparameterization Trick for symPareto**  To enable gradient-based learning, the VAE framework employs the reparameterization trick (Kingma & Welling, 2013). Exactly as Student's $t$-distribution can be represented in terms of Gaussian and chi-squared variables, symPareto admits a representation of a Laplace-Gamma mixture, allowing easy reparameterization.

**Proposition 3.1.** *Let* $Z \sim \mathcal{L}_n(0, I_n)$ *and* $W \sim \mathrm{Gamma}(\nu, 1)$ *be independent, where* $\mathcal{L}_n(0, I_n)$ *has i.i.d. univariate Laplace* $\mathcal{L}_1(0, 1)$ *components. Then* $T := (\nu/W)Z$ *satisfies* $T \sim \mathcal{P}_n(0, \mathbf{1}_n, \nu)$.

## 3.2 DERIVATION OF THE $\gamma$-LOSS

The computation of $\mathcal{D}_\gamma(q_\phi \,\|\, p_\theta)$ with $\gamma = -\frac{2}{2\nu+2m+n}$ results in a closed-form expression. By simplifying the constants and utilizing the *alternative* prior expression (cf. Kim et al. (2024)), we obtain the $\gamma$-loss function for the ParetoVAE under the mild condition $(m + 1)\gamma + 1 \neq 0$:

$$\mathcal{L}_\gamma^{\mathrm{alt}}(\theta, \phi) = \mathbb{E}_{x \sim p_{\mathrm{data}}} \left[ \frac{1}{2\sigma^2} \mathbb{E}_{z \sim q_\phi(\cdot|x)} \|x - \mu_\theta(z)\|_2^2 + \alpha \mathcal{D}_\gamma(q_\phi \,\|\, p_{\mathrm{alt}}) \right], \tag{9}$$

where $p_{\mathrm{alt}} = \mathcal{P}_m(0, k\mathbf{1}_m, \nu + n/2)$, $k = \frac{\nu}{\nu+2n} \left( \sigma^{-n}\pi^{-\frac{n}{2}} C_{n/2,\nu-1,\nu} \right)^{\frac{1}{\nu+n/2-1}}$, and $\alpha = -\frac{\gamma\nu}{2C_2}$ with

$$C_2 = \left\{ C_{m,\nu,\nu} T_{n,2\nu+2m} \sigma^{-n} \left( 2 + \frac{2m}{\nu} \right)^{\frac{n}{2}} \left( \frac{\nu - 1}{\nu + m + n/2 - 1} \right) \right\}^{\frac{\gamma}{1+\gamma}}.$$

In other words, the $\gamma$-loss can be understood as a sum of the mean squared error (MSE) and the $\gamma$-power divergence regularizer between the encoder and the *alternative* prior, $p_{\mathrm{alt}}$.

Moreover, to incorporate $\gamma$-flatness into the optimization process, we apply the result of Theorem 2.1 to the regularizer:

$$\mathcal{L}_\gamma^{\text{alt}}(\theta, \phi) \leq \mathbb{E}_{x \sim p_{\text{data}}} \left[ \frac{1}{2\sigma^2} \mathbb{E}_{z \sim q_\phi(\cdot|x)} \|x - \mu_\theta(z)\|_2^2 + \alpha \mathcal{D}_\gamma(q_{\phi,0} \,\|\, p_{\text{alt}}) + \alpha\beta \|\mu_\phi(x)\|_1 \right] =: \mathcal{L}_\gamma(\theta, \phi),$$
(10)

where $\beta$ is the value in Theorem 2.1 and $q_{\phi,0} = \mathcal{P}_m(z \mid 0, \sigma_\phi(x), \nu + n/2)$. Our practical objective $\mathcal{L}_\gamma(\theta, \phi)$ thus consists of three parts: the $\ell_2^2$-reconstruction loss, the $\gamma$-power divergence under $\gamma$-flatness, and an $\ell_1$ penalty term on $\mu_\phi(x)$.

**Limiting behavior**  We theoretically analyze the limiting behavior of $\alpha$, $k$, and $\beta$ in (9) as $\nu \to \infty$.

**Proposition 3.2** (Limiting behavior of the $\gamma$-loss). *For fixed $n$, $m$ and $x$, and $\gamma = -\frac{2}{2\nu + 2m + n}$,*

$$\lim_{\nu \to \infty} \alpha = \frac{1}{2}, \quad \lim_{\nu \to \infty} k = \lim_{\nu \to \infty} \beta = 1, \quad \lim_{\nu \to \infty} \mathcal{D}_\gamma(q_\phi \,\|\, p_{\text{alt}}) = \mathcal{D}_{KL}(q_{\phi,\infty} \,\|\, p_{\text{alt},\infty}),$$

*where $q_{\phi,\infty} = \mathcal{L}_m(\mu_\phi(x), \sigma_\phi(x))$ and $p_{\text{alt},\infty} = \mathcal{L}_m(0, \mathbf{1}_m)$ are limiting distributions of $q_\phi$ and $p_{\text{alt}}$.*

Proposition 3.2 shows that as $\nu \to \infty$, the $\gamma$-loss converges to the LaplaceVAE (LVAE) objective with the regularizer weight $\frac{1}{2}$; LVAE is a VAE with Laplace prior and encoder distributions together with a Gaussian decoder (Geadah et al., 2024). In this sense, ParetoVAE can be regarded as a heavy-tailed extension of the LVAE.

Moreover, in practice, the weight $\alpha$ and corresponding $\beta$ (coupled with $\alpha$) in (10) can be fine-tuned in the same spirit as the $\beta$-VAE (Higgins et al., 2017). Under such tuning, the modified $\gamma$-loss smoothly converges to the that of $\beta$-LVAE in the $\nu \to \infty$ limit.

## 3.3 Decoder Selection: SymPareto Decoder with $\ell_1$ Reconstruction Error

We may also modify the joint decoder distribution by replacing the $\ell_2^2$ reconstruction error term $\|x - \mu_\theta(z)\|_2^2$ in (8) with an $\ell_1$-norm-based variant:

$$p_\theta(x, z) \propto \left[ 1 + \frac{1}{\nu} \left( \|z\|_1 + \frac{\|x - \mu_\theta(z)\|_1}{\sigma} \right) \right]^{-(\nu + m + n)}.$$
(11)

This leads to the $\gamma$-loss function in which the MSE is replaced by the mean absolute error (MAE) when $\gamma = -\frac{1}{\nu + m + n}$:

$$\mathcal{L}_\gamma(\theta, \phi) = \mathbb{E}_{x \sim p_{\text{data}}} \left[ \frac{1}{\sigma} \mathbb{E}_{z \sim q_\phi(\cdot|x)} \|x - \mu_\theta(z)\|_1 + 2\alpha D_\gamma(q_{\phi,0} \,\|\, p_{\text{alt}}) + 2\alpha\beta \|\mu_\phi(x)\|_1 \right],$$
(12)

where $q_{\phi,0} = \mathcal{P}_m(0, \sigma_\phi(x), \nu + n)$, $p_{\text{alt}} = \mathcal{P}_m(0, k\mathbf{1}_m, \nu + n)$, and

$$k = \frac{\nu}{\nu + n} \left( C_{n, \nu-1, \nu} \sigma^{-n} \right)^{\frac{1}{\nu + n - 1}}, \quad \alpha = -\frac{\gamma\nu}{2C_2}, \quad C_2 = \left( C_{n+m,\nu,\nu} \frac{\nu - 1}{\nu + n + m - 1} \sigma^{-n} \right)^{\frac{\gamma}{1+\gamma}}.$$

Thanks to the MAE in the loss function, the model promotes robustness to extreme values. In the same way, we can apply a symPareto decoder to $t^3$VAE, thereby obtaining an MAE-based $t^3$VAE variant; see Appendix B.5.2.

## 4 Experiments

In this section, we evaluate ParetoVAE on both low and high-dimensional datasets under different decoder choices. We provide a comprehensive comparison against VAEs with different distributions: Gaussian VAE (VAE), LaplaceVAE (LVAE), and $t^3$VAE. Experiments details are given in Appendix C. We additionally report a wall-clock comparison of $\gamma$-loss training versus ELBO-based training; see Appendix C.4. Throughout, the hyperparameter $\nu$ is held fixed for $t^3$VAE and ParetoVAE (without tuning); see Appendix D for related discussion.

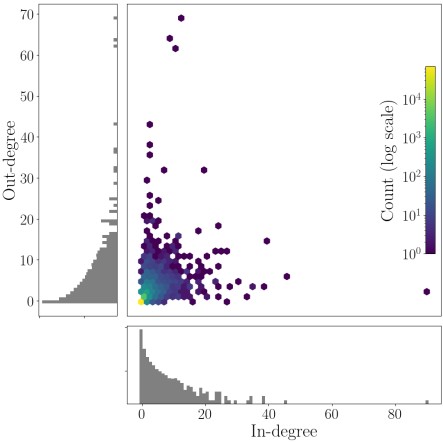

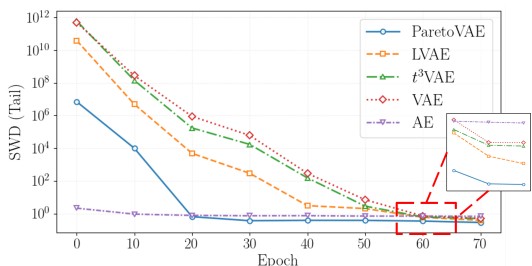

(b) SWD convergence curve on tail (log-scale)

| Model | SWD (Overall) (↓) | SWD (Tail) (↓) | $p$-value (Tail) |
|---|---|---|---|
| ParetoVAE | **0.044 ± 0.005** | **0.170 ± 0.029** | 0.221 ± 0.144 ✓ |
| LVAE | 0.055 ± 0.009 | 0.301 ± 0.084 | 0.119 ± 0.090 ✓ |
| $t^3$VAE | 0.055 ± 0.005 | 0.389 ± 0.040 | 0.181 ± 0.110 ✓ |
| VAE | 0.061 ± 0.018 | 0.402 ± 0.025 | 0.042 ± 0.030 ✗ |
| AE | 0.074 ± 0.030 | 0.621 ± 0.304 | 0.028 ± 0.027 ✗ |

(a) 2d hexbin of joint in/out-degree counts

(c) Overall SWD, tail SWD, and tail MMD $p$-values (mean ± standard deviation over 10 runs). Overall $p$-values are omitted since all models fail to reject.

Figure 3: Consolidated results for graph-degree reconstruction on the Epinions network.

## 4.1 SPARSE HEAVY-TAILED DATA ANALYSIS WITH $t$-DECODER

We evaluated ParetoVAE with a $t$-decoder on two representative tasks: low-dimensional graph degree distributions and high-dimensional word token vectors. As a point of comparison for tail modeling, we also include a deterministic autoencoder (AE) baseline. We employed the two-sample MMD test (Gretton et al., 2012) to test the hypothesis

$$H_0: p_{\text{data}} = p_{\text{recon}} \quad \text{vs.} \quad H_1: p_{\text{data}} \neq p_{\text{recon}} \tag{13}$$

and report the resulting $p$-values, where $p_{\text{recon}}$ is the distribution of the reconstructed samples.

**Graph degree reconstruction** We constructed a joint in- and out-degree count distribution of a directed graph using the SNAP Epinions social network (Richardson et al., 2003). In addition to the MMD test, we also measured the sliced 1-Wasserstein distance (SWD; Bonneel et al., 2014) in both head and tail regions, defined as the top 10% of samples according to their $\ell_2$ norm.

Figure 3a shows a hexagonal binned scatterplot (hexbin) of the joint degree counts, revealing a sparse, power–law shape. Figure 3b presents SWD convergence curves during training. We observe that the models with $\ell_1$-regularization (ParetoVAE, LVAE) converge more rapidly than those with $\ell_2^2$-regularization (VAE, $t^3$VAE), indicating enhanced robustness to extreme values. Figure 3c summarizes SWD metrics and tail MMD $p$-values. In all metrics, ParetoVAE achieved the lowest SWDs in both parts, showing its ability to capture tails and sparse extremes. By contrast, VAE and AE often rejected $H_0$, indicating poor tail fits.

**Word frequency analysis** Based on the WikiText-2 dataset (Merity et al., 2017), we constructed 19,962-dimensional bag-of-words representations from all tokens occurring at least 5 times. For head-tail analysis, the *head* was defined as the set of the 2,241 most frequent words, and the *tail* as the set of the 2,241 least frequent words among those with frequency $\geq 5$. For each part, we report the overlap ratio between the sets induced by reconstructions and the ground truth and the Jaccard similarity between the two sets.

As summarized in Table 2, ParetoVAE achieved the highest overlap and Jaccard scores for both head and tail. It also attains non-negligible MMD $p$-values on both parts, whereas baselines typically rejected $H_0$ in at least one part, indicating that ParetoVAE better captures the power-law structure. In contrast, AE captured neither tail nor head, and the others were only good at reconstructing head.

Table 2: Metrics for word frequency reconstruction on Wikitext-2. Reported values are the mean $\pm$ standard deviation over three runs. ✓ denotes cases where the $p$-value does not reject the null hypothesis at conventional significance levels (0.05 or 0.01), whereas ✗ indicates rejection.

| Model | Head | | | Tail | | |
|---|---|---|---|---|---|---|
| | Overlap (↑) | Jaccard (↑) | $p$-value | Overlap (%) | Jaccard (↑) | $p$-value |
| ParetoVAE | $\mathbf{0.981 \pm 0.001}$ | $\mathbf{0.964 \pm 0.001}$ | $0.417 \pm 0.237$ ✓ | $\mathbf{0.717 \pm 0.035}$ | $\mathbf{0.560 \pm 0.043}$ | $0.178 \pm 0.161$ ✓ |
| LVAE | $0.772 \pm 0.008$ | $0.629 \pm 0.010$ | $0.233 \pm 0.148$ ✓ | $0.230 \pm 0.003$ | $0.130 \pm 0.002$ | $0.001 \pm 0.000$ ✗ |
| $t^3$VAE | $0.739 \pm 0.002$ | $0.586 \pm 0.002$ | $0.665 \pm 0.116$ ✓ | $0.226 \pm 0.001$ | $0.127 \pm 0.001$ | $0.001 \pm 0.000$ ✗ |
| VAE | $0.775 \pm 0.017$ | $0.633 \pm 0.022$ | $0.229 \pm 0.200$ ✓ | $0.224 \pm 0.009$ | $0.126 \pm 0.006$ | $0.001 \pm 0.001$ ✗ |
| AE | $0.642 \pm 0.007$ | $0.473 \pm 0.008$ | $0.001 \pm 0.000$ ✗ | $0.197 \pm 0.004$ | $0.109 \pm 0.003$ | $0.001 \pm 0.000$ ✗ |

Table 3: Quantitative results for denoising task with a noise probability of 0.5 on various datasets. Mean $\pm$ standard deviation reported for 5 runs (class-related metrics are not reported for CelebA).

| Dataset | Model | PSNR (↑) | SSIM (↑) | Accuracy(↑) | Consistency (↑) |
|---|---|---|---|---|---|
| MNIST | ParetoVAE | $\mathbf{24.185 \pm 0.074}$ | $\mathbf{0.950 \pm 0.001}$ | $\mathbf{0.909 \pm 0.005}$ | $\mathbf{0.983 \pm 0.001}$ |
| | LVAE | $20.246 \pm 0.154$ | $0.891 \pm 0.025$ | $0.828 \pm 0.011$ | $0.954 \pm 0.005$ |
| | $t^3$VAE | $22.985 \pm 0.389$ | $0.935 \pm 0.005$ | $0.799 \pm 0.019$ | $0.979 \pm 0.001$ |
| | VAE | $18.516 \pm 0.138$ | $0.840 \pm 0.002$ | $0.741 \pm 0.004$ | $0.913 \pm 0.002$ |
| SVHN | ParetoVAE | $\mathbf{25.980 \pm 0.630}$ | $\mathbf{0.878 \pm 0.010}$ | $\mathbf{0.215 \pm 0.005}$ | $\mathbf{0.782 \pm 0.021}$ |
| | LVAE | $23.276 \pm 0.374$ | $0.786 \pm 0.018$ | $0.204 \pm 0.009$ | $0.528 \pm 0.057$ |
| | $t^3$VAE | $23.488 \pm 1.864$ | $0.797 \pm 0.076$ | $0.198 \pm 0.003$ | $0.570 \pm 0.208$ |
| | VAE | $21.028 \pm 0.309$ | $0.690 \pm 0.013$ | $0.197 \pm 0.002$ | $0.276 \pm 0.020$ |
| CIFAR10 | ParetoVAE | $\mathbf{20.552 \pm 0.119}$ | $\mathbf{0.723 \pm 0.002}$ | $\mathbf{0.368 \pm 0.002}$ | $\mathbf{0.341 \pm 0.006}$ |
| | LVAE | $16.622 \pm 0.105$ | $0.498 \pm 0.003$ | $0.285 \pm 0.008$ | $0.181 \pm 0.005$ |
| | $t^3$VAE | $19.584 \pm 0.064$ | $0.664 \pm 0.004$ | $0.316 \pm 0.001$ | $0.259 \pm 0.014$ |
| | VAE | $15.726 \pm 0.068$ | $0.457 \pm 0.006$ | $0.234 \pm 0.016$ | $0.165 \pm 0.005$ |
| Omniglot | ParetoVAE | $\mathbf{20.783 \pm 0.389}$ | $\mathbf{0.903 \pm 0.007}$ | $\mathbf{0.257 \pm 0.003}$ | $\mathbf{0.564 \pm 0.038}$ |
| | LVAE | $11.953 \pm 0.031$ | $0.716 \pm 0.001$ | $0.222 \pm 0.026$ | $0.000 \pm 0.000$ |
| | $t^3$VAE | $11.918 \pm 0.034$ | $0.712 \pm 0.001$ | $0.032 \pm 0.005$ | $0.000 \pm 0.000$ |
| | VAE | $11.917 \pm 0.035$ | $0.712 \pm 0.001$ | $0.033 \pm 0.002$ | $0.000 \pm 0.000$ |
| CelebA | ParetoVAE | $\mathbf{25.125 \pm 0.065}$ | $\mathbf{0.818 \pm 0.002}$ | - | - |
| | LVAE | $21.487 \pm 0.027$ | $0.708 \pm 0.003$ | - | - |
| | $t^3$VAE | $22.406 \pm 0.452$ | $0.741 \pm 0.012$ | - | - |
| | VAE | $18.554 \pm 0.017$ | $0.598 \pm 0.000$ | - | - |

## 4.2 Image Denoising Application with SymPareto Decoder

We conducted a denoising task using ParetoVAE with a symPareto decoder on benchmark datasets encompassing MNIST (Deng, 2012), SVHN (Netzer et al., 2011), Omniglot (Lake et al., 2015), and CelebA (Liu et al., 2015). Evaluation metrics were comprised of PSNR and SSIM to assess the quality of images reconstructed from corruption. For class-related metrics, we report: (1) $\overline{\text{accuracy}}$, obtained by applying linear probing directly to the latent variables sampled from the corrupted images. (2) (class) **consistency**, quantified as the agreement between the class predictions of a clean and reconstructed image using an pretrained external classifier exclusively on clean images.

Similarly to Vincent et al. (2008), we corrupted the original data $x$ to obtain noisy inputs $x'$, which are passed through the VAE models to reconstruct $\hat{x}$. The main objective includes an MAE loss between $x$ and $\hat{x}$. This procedure has been shown to yield a theoretically valid objective for VAEs (Im et al., 2016), and we extend this argument to ParetoVAE in Appendix C.5. The results are summarized in Table 3. Across all datasets, ParetoVAE consistently outperformed baseline models. We attribute this advantage to the MAE term in the $\gamma$-loss, which improves robustness to outliers and enhances recovery of fine-grained structures. The $t^3$VAE was competitive on MNIST but its performance degraded on complex datasets like Omniglot, highlighting the advantage of ParetoVAE in high-dimensional denoising.

Figure 4 compares denoised images from datasets corrupted with salt-and-pepper noise, applied either in RGB or grayscale. On CelebA (Figure 4a), $t^3$VAE yielded visually sharp reconstructions than VAE, but often generated samples not too close to the input. LVAE better preserved input-specific features,

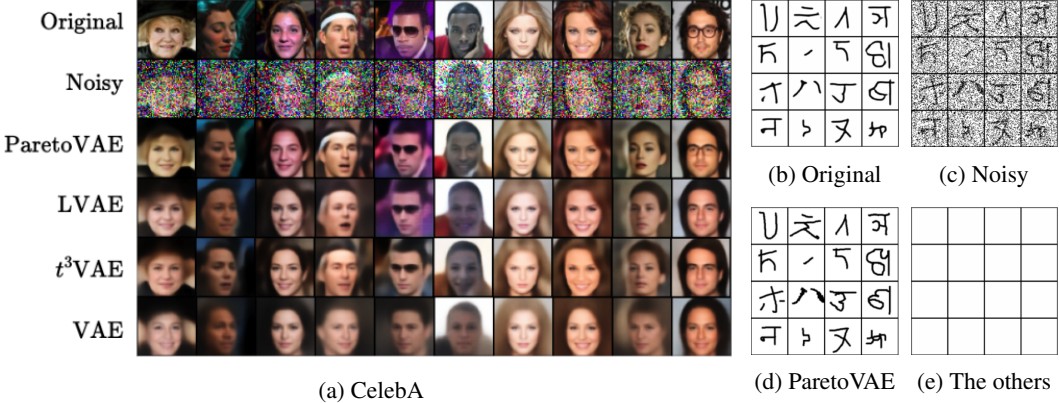

Figure 4: Denoising results on the benchmark datasets (with noise probability of 0.5). (a) displays the results on CelebA. For Omniglot, (b) shows the original images, (c) the noisy images, and the denoised images from ParetoVAE and other VAEs are shown in (d), and (e), respectively.

though with lower fidelity. ParetoVAE appears to combine both strengths, retaining distinctive features while achieving superior reconstruction quality, illustrating the benefit of the symPareto distribution.

Interestingly, on Omniglot, all models except ParetoVAE failed to reconstruct the original images from noisy inputs, even after fine-tuning hyperparameters (Figure 4d and Figure 4e). This outcome may stem from the fact that Omniglot is highly sparse in terms of classes, making it difficult for light-tailed or $\ell_2^2$-based regularization schemes to capture its underlying structure.

## 5 CONCLUSION

We have introduced the multivariate symmetric Pareto distribution (symPareto) and proposed Pareto-VAE, which employs symPareto prior and encoder with flexible decoders (Student's $t$ or symPareto). By leveraging the information-geometric $\gamma$-power divergence and the joint minimization viewpoint to variational inference, ParetoVAE induces a tractable $\gamma$-loss as an alternative to the ELBO-based one. Empirical results show that ParetoVAE captures the power-law behavior of the data and enhances robustness to outliers. Further discussion is provided in Appendix D.

## ACKNOWLEDGEMENTS

All authors were supported by the AI-Bio Research Grant through Seoul National University (No. 0413-20230050), Institute of Information & communications Technology Planning & Evaluation (IITP) grant funded by the Korea government (MSIT, No. RS-2021-II211343, Artificial Intelligence Graduate School Program (Seoul National University)) and by the National Research Foundation of Korea (NRF) grant funded by the Korea government (MSIT, No. RS-2024-00337691).

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

# Contents

# A  Some Properties of the Multivariate Symmetric Pareto Distributions

In this section, we provide some properties of the multivariate symmetric Pareto distributions.

## A.1  From Mardia's One-sided Pareto to the Symmetric Pareto via Reparameterization

We start from Mardia's $n$-variate Pareto distribution of the first kind (Mardia, 1962), whose joint density is given by

$$p_n(x_1, \ldots, x_n \mid \sigma, \nu) = \frac{\Gamma(\nu + n)}{\Gamma(\nu)} \frac{1}{\bar{\sigma}} \left( \sum_{i=1}^{n} \frac{x_i}{\sigma_i} - n + 1 \right)^{-(\nu+n)}, \qquad x_i > \sigma_i > 0, \ \nu > 0,$$

where $\bar{\sigma} = \prod_{i=1}^{n} \sigma_i$. This form has a lower-bounded support anchored at the vector $\sigma$.

To shift the support to the positive orthant with a unit offset, we reparameterize by $y_i = x_i - \sigma_i$. The density becomes

$$p_n(y_1, \ldots, y_n \mid \sigma, \nu) = \frac{\Gamma(\nu + n)}{\Gamma(\nu)} \frac{1}{\bar{\sigma}} \left( 1 + \sum_{i=1}^{n} \frac{y_i}{\sigma_i} \right)^{-(\nu+n)}, \qquad y_i > 0.$$

Next, to align this form with the $\gamma$-flat family in (4), we rescale the coordinate-wise scales by $\sigma_i \mapsto \nu \sigma_i$. This yields the one-sided (positive-orthant) Pareto density

$$\mathcal{P}_n^+(y_1, \ldots, y_n \mid \sigma, \nu) = \frac{\Gamma(\nu + n)}{\nu^n \Gamma(\nu)} \frac{1}{\bar{\sigma}} \left( 1 + \frac{1}{\nu} \sum_{i=1}^{n} \frac{y_i}{\sigma_i} \right)^{-(\nu+n)}, \qquad y_i > 0.$$

Finally, we obtain a *symmetric* distribution on the whole space by extending the support from the positive orthant to $\mathbb{R}^n$. Concretely, replace $y_i$ by $|x_i - \mu_i|$ (introducing a location parameter $\mu$) and adjust the normalizing constant by a factor $2^{-n}$ since the support is extended to all $2^n$ orthants. This gives the multivariate symmetric Pareto (symPareto) density

$$\mathcal{P}_n(x \mid \mu, \sigma, \nu) = \frac{C_{n,\nu,\nu}}{\bar{\sigma}} \left( 1 + \frac{1}{\nu} \left\| \frac{x - \mu}{\sigma} \right\|_1 \right)^{-(\nu+n)},$$

which coincides with (3). Here $C_{n,\nu_1,\nu_2} = \dfrac{\Gamma(\nu_1 + n)}{(2\nu_2)^n \Gamma(\nu_1)}$ and the division $(x - \mu)/\sigma$ is understood element-wise. In this way, the construction proceeds seamlessly from Mardia's one-sided form, through a reparameterization and $\nu\sigma$-scaling that reveals the $\gamma$-flat structure, to the desired symmetric extension on $\mathbb{R}^n$.

**Marginal distribution of the symPareto distribution**  For the $n$-variate standard symPareto distribution, the marginal distribution of each component $X_i$ is itself a univariate symPareto distribution. Its density can be computed as

$$
\begin{aligned}
p_1(x_1) &= \int_{\mathbb{R}} \cdots \int_{\mathbb{R}} C_{n,\nu,\nu} \left( 1 + \tfrac{1}{\nu} \|x\|_1 \right)^{-(\nu+n)} dx_2 \cdots dx_n \\
&= \frac{\Gamma(\nu + n)}{2\Gamma(\nu)\nu^n} \cdot \frac{\nu^{n-1}}{(\nu + n - 1) \cdots (\nu + 1)} \left( 1 + \tfrac{1}{\nu} |x_1| \right)^{-(\nu+1)} \\
&= \frac{1}{2} \left( 1 + \frac{1}{\nu} |x_1| \right)^{-(\nu+1)}.
\end{aligned}
$$

Thus each marginal has the same symmetric Pareto form in one dimension. By restricting to the positive half-line ($x > 0$) and writing $\nu = 1/\xi$, this reduces to

$$p(x) = (1 + \xi x)^{-(1/\xi + 1)}, \qquad x > 0,$$

which coincides with the generalized Pareto distribution (Pickands, 1975). This fact is directly connected to classical extreme-value theory. According to the Pickands–Balkema–De Haan theorem (Balkema & de Haan, 1974; Pickands, 1975), the GPD arises as the universal limit law for threshold exceedances of a broad class of distributions. Therefore, the symPareto distribution inherits marginal tail behavior that is theoretically consistent with the limit structure predicted by extreme-value theory.

### A.2 STATISTICS OF THE SYMPARETO DISTRIBUTION

**Mean and variance** Let $X \sim \mathcal{P}_1(0, 1, \nu)$. The expectation exists for $\nu > 1$, and by symmetry of the density we have $E[X] = 0$. When $\nu > 2$, the variance is finite and can be computed as

$$Var(X) = E[X^2]$$

$$= \int_{-\infty}^{\infty} \frac{x^2}{2} \left(1 + \frac{1}{\nu}|x|\right)^{-(\nu+1)} dx$$

$$= \int_{-\infty}^{0} \frac{x^2}{2} \left(1 - \frac{1}{\nu}x\right)^{-(\nu+1)} dx + \int_{0}^{\infty} \frac{x^2}{2} \left(1 + \frac{1}{\nu}x\right)^{-(\nu+1)} dx$$

$$= \frac{\nu^3}{2} \left(\int_{1}^{\infty} (1-y)^2 y^{-(\nu+1)} \, dy + \int_{1}^{\infty} (z-1)^2 z^{-(\nu+1)} \, dz\right) \quad \left(y = 1 - \frac{1}{\nu}x, \; z = 1 + \frac{1}{\nu}x\right)$$

$$= \nu^3 \left(\frac{1}{\nu - 2} - \frac{2}{\nu - 1} + \frac{1}{\nu}\right) = \frac{2\nu^2}{(\nu - 1)(\nu - 2)}.$$

For the multivariate case $X \sim \mathcal{P}_n(0, \sigma, \nu)$, the covariance between distinct components also vanishes by spherical symmetry (Fang et al., 2017), although this does not imply independence.

**The first central/non-central absolute moment** Let $X \sim \mathcal{P}_1(0, 1, \nu)$. For $\nu > 1$, the first central absolute moment of $X$ is

$$E[|X|] = \int_{0}^{\infty} x \left(1 + \frac{1}{\nu}x\right)^{-(\nu+1)} dx$$

$$= \nu^2 \cdot \int_{0}^{\infty} t \, (1 + t)^{-(\nu+1)} \, dt = \frac{\nu}{\nu - 1}.$$

The first non-central absolute moment about $t \in \mathbb{R}$ is given by

$$E[|X + t|] = |t| + \frac{\nu}{\nu - 1} \left(1 + \frac{|t|}{\nu}\right)^{-\nu+1}. \tag{14}$$

To compute this, we split the integral into two cases depending on the sign of $t$:

$$E[|X + t|] = \int_{-\infty}^{\infty} \frac{|x + t|}{2} \left(1 + \frac{1}{\nu}|x|\right)^{-(\nu+1)} dx$$

$$= \underbrace{\int_{-\infty}^{-t} -\frac{(x+t)}{2} \left(1 + \frac{1}{\nu}|x|\right)^{-(\nu+1)} dx}_{I_1} + \underbrace{\int_{-t}^{\infty} \frac{x+t}{2} \left(1 + \frac{1}{\nu}|x|\right)^{-(\nu+1)} dx}_{I_2}$$

1) $t \geq 0$

$$I_1 = -\frac{1}{2} \int_{-\infty}^{-t} (x+t) \left(1 - \frac{1}{\nu}x\right)^{-(\nu+1)} dx$$

$$= -\frac{1}{2} \int_{1+t/\nu}^{\infty} (\nu(1-y) + t)y^{-(\nu+1)} \, \nu dy$$

$$= -\frac{\nu + t}{2} \left(1 + \frac{t}{\nu}\right)^{-\nu} + \frac{\nu^2}{2(\nu - 1)} \left(1 + \frac{t}{\nu}\right)^{-\nu+1}$$

$$= \frac{\nu}{2(\nu - 1)} \left(1 + \frac{t}{\nu}\right)^{-\nu+1}.$$

$$I_2 = \frac{1}{2} \int_{-t}^{0} (x+t) \left(1 - \frac{1}{\nu}x\right)^{-(\nu+1)} dx + \frac{1}{2} \int_{0}^{\infty} (x+t) \left(1 + \frac{1}{\nu}x\right)^{-(\nu+1)} dx$$

$$= \frac{1}{2} \int_{1}^{1+t/\nu} (\nu(1-y)+t)y^{-(\nu+1)} \nu dy + \frac{1}{2} \int_{1}^{\infty} (\nu(z-1)+t)z^{-(\nu+1)}\nu dz$$

$$= \frac{\nu+t}{2} \left\{ 1 - \left(1 + \frac{t}{\nu}\right)^{-\nu} \right\} - \frac{\nu^2}{2(\nu-1)} \left\{ 1 - \left(1 + \frac{t}{\nu}\right)^{-\nu+1} \right\} + \frac{\nu^2}{2(\nu-1)} + \frac{t-\nu}{2}$$

$$= t + \frac{\nu}{2(\nu-1)} \left(1 + \frac{t}{\nu}\right)^{-\nu+1}$$

$$\therefore E[|X+t|] = I_1 + I_2 = t + \frac{\nu}{\nu-1} \left(1 + \frac{t}{\nu}\right)^{-\nu+1}, \ t \geq 0.$$

2) $t < 0$

$$I_1 = -\frac{1}{2} \int_{-\infty}^{0} (x+t) \left(1 - \frac{1}{\nu}x\right)^{-(\nu+1)} dx - \frac{1}{2} \int_{0}^{-t} (x+t) \left(1 + \frac{1}{\nu}x\right)^{-(\nu+1)} dx$$

$$= -\frac{1}{2} \int_{1}^{\infty} (\nu(1-y)+t)y^{-(\nu+1)} \nu dy - \frac{1}{2} \int_{1}^{1-t/\nu} (\nu(z-1)+t)z^{-(\nu+1)}\nu dz$$

$$= -\frac{(\nu+t)}{2} + \frac{\nu^2}{2(\nu-1)} - \frac{t-\nu}{2} \left\{ 1 - \left(1 - \frac{t}{\nu}\right)^{-\nu} \right\} - \frac{\nu^2}{2(\nu-1)} \left\{ 1 - \left(1 - \frac{t}{\nu}\right)^{-\nu+1} \right\}$$

$$= -t + \frac{\nu}{2(\nu-1)} \left(1 - \frac{t}{\nu}\right)^{-\nu+1}$$

$$I_2 = \frac{1}{2} \int_{-t}^{\infty} (x+t) \left(1 + \frac{1}{\nu}x\right)^{-(\nu+1)} dx$$

$$= \frac{1}{2} \int_{1-t/\nu}^{\infty} (\nu(z-1)+t)z^{-(\nu+1)} \nu dz$$

$$= \frac{t-\nu}{2} \left(1 - \frac{t}{\nu}\right)^{-\nu} + \frac{\nu^2}{2(\nu-1)} \left(1 - \frac{t}{\nu}\right)^{-\nu+1}$$

$$= \frac{\nu}{2(\nu-1)} \left(1 - \frac{t}{\nu}\right)^{-\nu+1}.$$

$$\therefore E[|X+t|] = I_1 + I_2 = -t + \frac{\nu}{\nu-1} \left(1 - \frac{t}{\nu}\right)^{-\nu+1}, \ t < 0.$$

Combining both cases leads to the unified expression in (14).

A.3 CONVERGENCE IN DISTRIBUTION OF THE SYMPARETO

We now investigate convergence in distribution of the symPareto with respect to the parameter $\nu$. Notice that according to Scheffé's theorem (Scheffe, 1947), it's enough to check the convergence of the probability density functions as assuming regular condition for the measure space.

**Theorem A.1** (Convergence in distribution of the symPareto under $\nu$ goes infinity). *For $n \geq 1$, fix $\mu, x \in \mathbb{R}^n$, and $\sigma \in \mathbb{R}^n_+$. Then, the limiting distribution of $\mathcal{P}_n(x \mid \mu, \sigma, \nu)$ is given by:*

$$\mathcal{P}_n(x \mid \mu, \sigma, \nu) \xrightarrow{d} L_n(x \mid \mu, \sigma), \tag{15}$$

*where $\mathcal{L}_n(x \mid \mu, \sigma)$ has i.i.d. univariate Laplace $\mathcal{L}_1(\mu_i, \sigma_i)$ components, which is then given by*

$$\mathcal{L}_n(x \mid \mu, \sigma) = \frac{1}{2^n \bar{\sigma}} \exp \left( - \left\| \frac{x-\mu}{\sigma} \right\|_1 \right).$$

*Proof.* We analyze the limit of the normalizing constant $C_{n,\nu,\nu}$ as $\nu \to \infty$. Using the Stirling's approximation,

$$\Gamma(x+1) \sim \sqrt{2\pi x}\left(\frac{x}{e}\right)^x, \quad x > 0,$$

we obtain

$$\begin{aligned}
\lim_{\nu\to\infty} C_{n,\nu,\nu} &= \frac{1}{2^n} \lim_{\nu\to\infty} \frac{\Gamma(\nu+1)}{\nu^n \Gamma(\nu)} \\
&= \frac{1}{2^n} \lim_{\nu\to\infty} \sqrt{\frac{\nu+n-1}{\nu-1}} \frac{\left(1+\frac{n}{\nu-1}\right)^{\nu-1}\left(\frac{\nu+n-1}{e}\right)^n}{\nu^n} \\
&= \frac{1}{2^n}.
\end{aligned} \tag{16}$$

For the unnormalized part (kernel) of the density, we observe that

$$\begin{aligned}
\lim_{\nu\to\infty} \frac{1}{\bar{\sigma}}\left(1+\frac{1}{\nu}\left\|\frac{x-\mu}{\sigma}\right\|_1\right)^{-(\nu+n)} &= \frac{1}{\bar{\sigma}}\exp\left(-\left\|\frac{x-\mu}{\sigma}\right\|_1\right) \\
&= \prod_{i=1}^{n} \frac{1}{\sigma_i}\exp\left(-\left|\frac{x_i-\mu_i}{\sigma_i}\right|\right),
\end{aligned}$$

which shows that the symPareto converges pointwisely to the product of univariate Laplace. □

### A.4 PROOF OF THEOREM 2.1

*Proof.* Consider two $n$-variate symPareto distributions $p = \mathcal{P}_n(\mu_1, \sigma_1, \nu)$ and $q = \mathcal{P}_n(\mu_2, \sigma_2, \nu)$. To analyze these distributions within a $\gamma$-flat manifold, we set $\gamma = -\frac{1}{\nu+n}$. We then compute the integration for $\gamma$-power entropy and $\gamma$-power cross-entropy terms as follows:

$$\begin{aligned}
\int p(x)^{1+\gamma} dx &= C_{n,\nu,\nu}^{1+\gamma} \bar{\sigma}_1^{-\gamma} \cdot \bar{\sigma}_1^{-1} \int \left(1+\frac{1}{\nu}\left\|\frac{x-\mu_1}{\sigma_1}\right\|_1\right)^{-(\nu-1+n)} dx \\
&= \bar{\sigma}_1^{-\gamma} \frac{C_{n,\nu,\nu}^{1+\gamma}}{C_{n,\nu-1,\nu}} \\
&= \bar{\sigma}_1^{-\gamma} C_{n,\nu,\nu}^{\gamma} \left(\frac{\nu+n-1}{\nu-1}\right). \\
\int q(x)^{1+\gamma} dx &= \bar{\sigma}_2^{-\gamma} C_{n,\nu,\nu}^{\gamma}\left(\frac{\nu+n-1}{\nu-1}\right),
\end{aligned}$$

since

$$\frac{C_{n,\nu,\nu}}{C_{n,\nu-1,\nu}} = \frac{\frac{\Gamma(\nu+n)}{\Gamma(\nu)(2\nu)^n}}{\frac{\Gamma(\nu+n-1)}{\Gamma(\nu-1)(2\nu)^n}} = \frac{\nu+n-1}{\nu-1}. \tag{17}$$

Also,

$$\begin{aligned}
\int p(x)q(x)^\gamma dx &= C_{n,\nu,\nu}^{\gamma} \bar{\sigma}_2^{-\gamma} \mathbb{E}_{x\sim p}\left[\left(1+\frac{1}{\nu}\left\|\frac{x-\mu_2}{\sigma_2}\right\|_1\right)\right] \\
&= C_{n,\nu,\nu}^{\gamma} \bar{\sigma}_2^{-\gamma}\left(1+\frac{1}{\nu}\sum_{i=1}^{n}\mathbb{E}_{x\sim p}\left[\left|\frac{x_i-\mu_{2,i}}{\sigma_{2,i}}\right|\right]\right) \\
&= C_{n,\nu,\nu}^{\gamma} \bar{\sigma}_2^{-\gamma}\left(1+\frac{1}{\nu}\sum_{i=1}^{n}\frac{\sigma_{1,i}}{\sigma_{2,i}}\mathbb{E}_{z\sim\mathcal{P}(0,\mathbf{1}_n,\nu)}\left[\left|z+\frac{\mu_{\Delta,i}}{\sigma_{1,i}}\right|\right]\right), \quad (\mu_{\Delta,i} := \mu_{1,i}-\mu_{2,i}) \\
&= C_{n,\nu,\nu}^{\gamma} \bar{\sigma}_2^{-\gamma}\left(1+\frac{1}{\nu}\sum_{i=1}^{n}\frac{|\mu_{\Delta,i}|}{\sigma_{2,i}}+\frac{\nu}{\nu-1}\frac{\sigma_{1,i}}{\sigma_{2,i}}\left(1+\frac{1}{\nu}\frac{|\mu_{\Delta,i}|}{\sigma_{1,i}}\right)^{-\nu+1}\right) \quad (\because \text{Equation (14)}) \\
&= C_{n,\nu,\nu}^{\gamma} \bar{\sigma}_2^{-\gamma}\left(1+\frac{1}{\nu}\left\|\frac{\mu_\Delta}{\sigma_2}\right\|_1+\frac{1}{\nu-1}\left\|\frac{\sigma_1}{\sigma_2}\left(1+\frac{1}{\nu}\frac{|\mu_\Delta|}{\sigma_1}\right)^{-\nu+1}\right\|_1\right)
\end{aligned}$$

Hence, the $\gamma$-power entropy and cross-entropy are:

$$\mathcal{H}_\gamma(p) = -\left(\int p(x)^{1+\gamma}\right)^{\frac{1}{1+\gamma}} = -\bar{\sigma}_1^{-\frac{\gamma}{1+\gamma}} C_{n,\nu,\nu}^{\frac{\gamma}{1+\gamma}} \left(\frac{\nu+n-1}{\nu-1}\right)^{\frac{1}{1+\gamma}},$$

$$\mathcal{C}_\gamma(p,q) = -\left(\int p(x)q(x)^\gamma\right)\left(\int q(x)^{1+\gamma}\right)^{-\frac{\gamma}{1+\gamma}}$$

$$= -C_{n,\nu-1,\nu}^{\frac{\gamma}{1+\gamma}} \bar{\sigma}_2^{-\frac{\gamma}{1+\gamma}} \left(1 + \frac{1}{\nu}\left\|\frac{\mu_\Delta}{\sigma_2}\right\|_1 + \frac{1}{\nu-1}\left\|\frac{\sigma_1}{\sigma_2}\left(1+\frac{1}{\nu}\frac{|\mu_\Delta|}{\sigma_1}\right)^{-\nu+1}\right\|_1\right).$$

Combining the two terms, the $\gamma$-power divergence between $p$ and $q$ yields a closed-form expression:

$$\mathcal{D}_\gamma(p\,\|\,q) = \frac{1}{\gamma}\mathcal{C}_\gamma(p,q) - \frac{1}{\gamma}\mathcal{H}_\gamma(p)$$

$$= C'\left[\bar{\sigma}_2^{-\frac{\gamma}{1+\gamma}}\left(1 + \frac{1}{\nu}\left\|\frac{\mu_\Delta}{\sigma_2}\right\|_1 + \frac{1}{\nu-1}\left\|\frac{\sigma_1}{\sigma_2}\left(1+\frac{1}{\nu}\frac{|\mu_\Delta|}{\sigma_1}\right)^{-\nu+1}\right\|_1\right) - \bar{\sigma}_1^{-\frac{\gamma}{1+\gamma}}\left(\frac{\nu+n-1}{\nu-1}\right)\right],$$

$$(18)$$

where $C' := C'(n,\nu,\gamma) = (\nu+n)C_{n,\nu-1,\nu}^{\frac{\gamma}{1+\gamma}}$ and (17) is used in the last equation. We then put $\mu_\Delta = 0$ and get the closed form of $\mathcal{D}_\gamma(p_0\,\|\,q_0)$ (5).

Next, let $p_0 = \mathcal{P}_n(0,\sigma_1,\nu)$ and $q_0 = \mathcal{P}_n(0,\sigma_2,\nu)$. To obtain an upper bound of $\mathcal{D}_\gamma(p\,\|\,q)$ in terms of $\mathcal{D}_\gamma(p_0\,\|\,q_0)$, we apply the triangle inequality to $|x_i - \mu_{2,i}| \le |x_i - \mu_{1,i}| + |\mu_{1,i} - \mu_{2,i}|$. As a result, the cross-entropy term admits the following bound:

$$\int p(x)q(x)^\gamma = C_{n,\nu,\nu}^\gamma \bar{\sigma}_2^{-\gamma}\left(1 + \frac{1}{\nu}\sum_{i=1}^n \mathbb{E}_{x\sim p}\left[\left|\frac{x_i-\mu_{2,i}}{\sigma_{2,i}}\right|\right]\right)$$

$$\le C_{n,\nu,\nu}^\gamma \bar{\sigma}_2^{-\gamma}\left(1 + \frac{1}{\nu}\sum_{i=1}^n \mathbb{E}_{x\sim p}\left[\frac{\sigma_{1,i}}{\sigma_{2,i}}\left|\frac{x_i-\mu_{1,i}}{\sigma_{1,i}}\right| + \left|\frac{\mu_{1,i}-\mu_{2,i}}{\sigma_{2,i}}\right|\right]\right)$$

$$= C_{n,\nu,\nu}^\gamma \bar{\sigma}_2^{-\gamma}\left(1 + \frac{1}{\nu}\sum_{i=1}^n \frac{\sigma_{1,i}}{\sigma_{2,i}}\left(\frac{\nu}{\nu-1}\right) + \mathbb{E}_{x\sim p}\left[\left|\frac{\mu_{1,i}-\mu_{2,i}}{\sigma_{2,i}}\right|\right]\right)$$

$$= C_{n,\nu,\nu}^\gamma \bar{\sigma}_2^{-\gamma}\left(1 + \frac{1}{\nu-1}\left\|\frac{\sigma_1}{\sigma_2}\right\|_1 + \frac{1}{\nu}\left\|\frac{\mu_1-\mu_2}{\sigma_2}\right\|_1\right)$$

Using this result, we have an upper bound:

$$\mathcal{D}_\gamma(p\,\|\,q) \le C'\left[\bar{\sigma}_2^{-\frac{\gamma}{1+\gamma}}\left(1 + \frac{1}{\nu-1}\left\|\frac{\sigma_1}{\sigma_2}\right\|_1 + \frac{1}{\nu}\left\|\frac{\mu_1-\mu_2}{\sigma_2}\right\|_1\right) - \bar{\sigma}_1^{-\frac{\gamma}{1+\gamma}}\left(\frac{\nu+n-1}{\nu-1}\right)\right],$$

$$\le \mathcal{D}_\gamma(p_0\,\|\,q_0) + \left(\frac{C_{n,\nu-1,\nu}}{\bar{\sigma}_2}\right)^{\frac{\gamma}{1+\gamma}}\left(1 + \frac{n}{\nu}\right)\left\|\frac{\mu_1-\mu_2}{\sigma_2}\right\|_1$$

and equality holds when $\mu_1 = \mu_2$. $\qquad\square$

## B  COMPUTATION FOR PARETOVAE

This section presents details of computation involved with the ParetoVAE models. Appendix B.1 provides the proof of Proposition 3.1. The subsequent subsections describe the construction process for heavy-tailed VAE variants, including ParetoVAE with $t$-decoder, ParetoVAE with symPareto decoder, and $t^3$VAE with symPareto decoder.

### B.1  PROOF OF PROPOSITION 3.1

*Proof.* Let $Z \in \mathbb{R}^n$ and $W \in \mathbb{R}$ be independent random variables such that:

$$Z = (Z_1, \cdots, Z_n),\ Z_i \overset{\text{i.i.d.}}{\sim} \mathcal{L}_1(0,1),\quad W \sim \text{Gamma}(\nu,1),$$

and denote their probability density function as $f_Z(z)$ and $f_W(w)$:

$$f_W(w) = \frac{1}{\Gamma(\nu)} w^{\nu-1} e^{-w}, \quad w > 0, \qquad f_Z(z) = \frac{1}{2^n} \exp\left(-\|z\|_1\right), \quad z \in \mathbb{R}^n.$$

Let $S = \frac{Z}{W}$. Then the probability density function of $S$ is given by:

$$\begin{aligned}
f_S(s) &= \int_0^\infty f_{S,W}(s,w) dw \\
&= \int_0^\infty f_{Z,W}(sw, w) \cdot w^n \, dw \\
&= \int_0^\infty f_Z(sw) f_W(w) \cdot w^n \, dw \quad (\because Z \perp W) \\
&= \int_0^\infty w^n \cdot \left(\frac{1}{2} \exp\left(-\|sw\|_1\right)\right) \cdot \left(\frac{1}{\Gamma(\nu)} w^{\nu-1} \exp(-w)\right) dw \\
&= \frac{1}{2^n \Gamma(\nu)} \int_0^\infty w^{\nu+n-1} \cdot \exp(-w(\|s\|_1 + 1)) dw \\
&= \frac{\Gamma(\nu+n)}{2^n \Gamma(\nu)} (\|s\|_1 + 1)^{-(\nu+n)}.
\end{aligned}$$

Therefore, the probability density function of $T = \nu S$ is

$$f_T(t) = \frac{\Gamma(\nu+n)}{(2\nu)^n \Gamma(\nu)} \left(1 + \frac{\|t\|_1}{\nu}\right)^{-(\nu+n)} = \mathcal{P}_n(t \mid 0, \mathbf{1}_n, \nu).$$

$\square$

**Sampling algorithm**  From Proposition 3.1, we summarize the sampling algorithm for symPareto distribution from Laplace and Gamma distribution as Algorithm 1:

---
**Algorithm 1** Sampling algorithm for the symmetric multivariate Pareto distribution

---
**Require:** $N$: # of samples, $m$ : Latent dimension, $\nu$ : Hyperparameter for the Pareto prior
**Ensure:** $m$-variate symPareto sample $z \in \mathbb{R}^m$
  Sample $x_i \overset{\text{i.i.d.}}{\sim} \text{Laplace}(0,1)$, for $i = 1, \cdots, m$.
  Assign $x = (x_1, \ldots, x_n)$.
  Sample $w \sim \text{Gamma}(\nu, 1)$.
  Assign $z = x \cdot (\nu/w)$.

---

### B.2 GENERAL DECODER CONSTRUCTION: STUDENT'S $t$ AND SYMPARETO

Following Table 1, we provide an explicit construction of the joint decoder distributions followed by heavy-tailed VAE variants.

$$p_\theta(x, z) \propto \left[1 + \frac{1}{\nu}\left(\|z\|_p^p + \frac{1}{\sigma^q}\|x - \mu_\theta(z)\|_q^q\right)\right]^{-\left(\frac{\nu+m}{p} + \frac{n}{q}\right)}, \qquad p, q \in \{1, 2\}.$$

The prior can be generalized by

$$p(z) \propto \left(1 + \frac{1}{\nu}\|z\|_p^p\right)^{-\frac{\nu+m}{p}}, \qquad p \in \{1, 2\}.$$

which implies the symPareto prior ($p = 1$) or $t$-prior ($p = 2$). From this prior, the conditional decoder distribution can be derived as follows:

$$p_\theta(x \mid z) \propto \frac{\left[1 + \frac{1}{\nu}\left(\|z\|_p^p + \frac{1}{\sigma^q}\|x - \mu_\theta(z)\|_q^q\right)\right]^{-\left(\frac{\nu+m}{p} + \frac{n}{q}\right)}}{\left(1 + \frac{1}{\nu}\|z\|_p^p\right)^{-\frac{\nu+m}{p}}}$$

$$\propto \left[1 + \frac{1}{q/p(\nu+m)} \cdot \frac{q/p(\nu+m)}{\nu + \|z\|_p^p} \frac{1}{\sigma^q}\|x - \mu_\theta(z)\|_q^q\right]^{-\left(\frac{\nu+m}{p} + \frac{n}{q}\right)}.$$

By simplifying the ratio of the joint and the prior kernels, we obtain the following four cases depending on the choice of $p_\theta(x \mid z)$:

$$p_\theta(x \mid z) \propto \begin{cases} \mathcal{P}_n\left(x \mid \mu_\theta(z), \frac{\nu+\|z\|_1}{\nu+m}\sigma\mathbf{1}_n, \nu+m\right), & (p,q) = (1,1) \\ t_n\left(x \mid \mu_\theta(z), \frac{\nu+\|z\|_1}{2\nu+2m}\sigma^2 I_n, 2\nu+2m\right), & (p,q) = (1,2) \\ \mathcal{P}_n\left(x \mid \mu_\theta(z), \frac{2(\nu+\|z\|_2^2)}{\nu+m}\sigma\mathbf{1}_n, \frac{\nu+m}{2}\right), & (p,q) = (2,1) \\ t_n\left(x \mid \mu_\theta(z), \frac{\nu+\|z\|_2^2}{\nu+m}\sigma^2 I_n, \nu+m\right), & (p,q) = (2,2) \end{cases}$$

which implies symPareto decoder ($q = 1$) and $t$-decoder ($q = 2$). Note that the symPareto decoders employ element-wise scale parameters ($\sigma\mathbf{1}_n$), whereas $t$-decoders use covariance scaling ($\sigma^2 I_n$).

### B.3 Derivation of the $\gamma$-loss for ParetoVAE with $t$-decoder

#### B.3.1 Derivation of the Raw $\gamma$-loss

The ParetoVAE starts from the heavy-tailed joint decoder model:

$$p_\theta(x, z) \propto \left[1 + \frac{1}{\nu}\left(\|z\|_1 + \frac{\|x - \mu_\theta(z)\|_2^2}{\sigma^2}\right)\right]^{-(\nu+m+\frac{n}{2})}, \quad \sigma \in \mathbb{R}^+, \nu > 1.$$

We first compute the normalizing constant $C$ by separating the integration over $x$ and $z$:

$$\int_{\mathbb{R}^m}\int_{\mathbb{R}^n} p_\theta(x, z)\, dx dz$$

$$= \int_{\mathbb{R}^m}\left(1 + \frac{1}{\nu}\|z\|_1\right)^{-(\nu+m+\frac{n}{2})}\int_{\mathbb{R}^n}\left[1 + \frac{1}{2(\nu+m)}\frac{2(\nu+m)}{\nu+\|z\|_1}\frac{1}{\sigma^2}\|x - \mu_\theta(z)\|_2^2\right]^{-\frac{2\nu+2m+n}{2}} dx dz$$

$$= T_{n,2\nu+2m}^{-1}\sigma^n\left(2 + \frac{2m}{\nu}\right)^{-\frac{n}{2}}\int_{\mathbb{R}^m}\left(1 + \frac{1}{\nu}\|z\|_1\right)^{-(\nu+m)} dz$$

$$= T_{n,2\nu+2m}^{-1}\sigma^n\left(2 + \frac{2m}{\nu}\right)^{-\frac{n}{2}} C_{m,\nu,\nu}^{-1}$$

Thus, the normalizing constant $C$ is

$$C = C_{m,\nu,\nu}T_{n,2\nu+2m}\sigma^{-n}\left(2 + \frac{2m}{\nu}\right)^{\frac{n}{2}}.$$

Remark that $T_{n,2\nu+2m}$ is the normalizing constant of $t_n(0, I_n 2\nu+2m)$. This decomposition naturally yields the prior and conditional decoder distributions:

$$p(z) = \mathcal{P}_m(z \mid 0, \mathbf{1}_m, \nu)$$

$$p_\theta(x \mid z) = t_n(x \mid \mu_\theta(z), \frac{\nu+\|z\|_1}{2\nu+2m}\sigma^2 I_n, 2\nu+2m)$$

We also define the encoder as a symPareto distribution with $\nu + n/2$ to align the tail order with the joint exponent $(\nu + m + n/2)$:

$$q_\phi(z|x) = \mathcal{P}_m(z \mid \mu_\phi(x), \sigma_\phi(x), \nu + n/2).$$

Next, we compute the $\gamma$-power divergence between two joint distributions $q_\phi$ and $p_\theta$. Setting $\gamma = -\frac{2}{2\nu + 2m + n}$, the $\gamma$-entropy and $\gamma$-cross entropy corresponding to the joint distributions $p_\theta$ and $q_\phi$ can be expressed as follows:

$$
\begin{aligned}
\iint p_\theta(x,z)^{1+\gamma} dx dz &= \mathbb{E}_{z \sim p_Z} \mathbb{E}_{x \sim p_\theta(\cdot|z)} \left[ p_\theta(x,z)^\gamma \right] \\
&= C^\gamma \mathbb{E}_{z \sim p_Z} \mathbb{E}_{x \sim p_\theta(\cdot|z)} \left[ 1 + \frac{1}{\nu} \left( \|z\|_1 + \frac{1}{\sigma^2} \|x - \mu_\theta(z)\|_2^2 \right) \right] \\
&= C^\gamma \mathbb{E}_{z \sim p_Z} \left[ 1 + \frac{1}{\nu} \|z\|_1 + \frac{1}{\nu\sigma^2} \mathrm{Tr} \left( \frac{2\nu + 2m}{2\nu + 2m - 2} \cdot \frac{\nu + \|z\|_1}{2\nu + 2m} \sigma^2 I_n \right) \right] \\
&= C^\gamma \mathbb{E}_{z \sim p_Z} \left[ \left( 1 + \frac{n}{2\nu + 2m - 2} \right) \left( 1 + \frac{1}{\nu} \|z\|_1 \right) \right] \\
&= C^\gamma \left( 1 + \frac{n/2}{\nu + m - 1} \right) \left( 1 + \frac{m}{\nu - 1} \right) \\
&= C^\gamma \left( \frac{\nu + m + n/2 - 1}{\nu - 1} \right), \quad \nu > 1
\end{aligned}
$$

and

$$
\begin{aligned}
\iint q_\phi(x,z) p_\theta(x,z)^\gamma dx dz &= \mathbb{E}_{x \sim p_{\mathrm{data}}} \mathbb{E}_{z \sim q_\phi(\cdot|x)} \left[ p_\theta(x,z)^\gamma \right] \\
&= C^\gamma \mathbb{E}_{x \sim p_{\mathrm{data}}} \mathbb{E}_{z \sim q_\phi(\cdot|x)} \left[ 1 + \frac{1}{\nu} \left( \|z\|_1 + \frac{1}{\sigma^2} \|x - \mu_\theta(z)\|_2^2 \right) \right] \\
&= C^\gamma \mathbb{E}_{x \sim p_{\mathrm{data}}} \left[ 1 + \frac{1}{\nu} \|\mu_\phi(x)\|_1 + \frac{\nu + n/2}{\nu(\nu + n/2 - 1)} \|r_\phi(x)\|_1 + \frac{1}{\nu\sigma^2} \mathbb{E}_{z \sim q_\phi(\cdot|x)} \|x - \mu_\theta(z)\|_2^2 \right],
\end{aligned}
$$

where

$$r_\phi(x) = \sigma_\phi(x) \left( 1 + \frac{|\mu_\phi(x)|}{(\nu + n/2)\sigma_\phi(x)} \right)^{-(\nu + n/2 - 1)}.$$

Thus, the $\gamma$-power entropy becomes

$$
\begin{aligned}
\mathcal{H}_\gamma(q_\phi) &= -\left( \iint q_\phi(x,z)^{1+\gamma} dx dz \right)^{\frac{1}{1+\gamma}} \\
&= -\left( \int \left( \int q_\phi(z|x)^{1+\gamma} dz \right) p_{\mathrm{data}}(x)^{1+\gamma} dx \right)^{\frac{1}{1+\gamma}} \\
&= -\underbrace{C_{m,\nu+n,\nu+n}^{\frac{\gamma}{1+\gamma}} \left( \frac{\nu + m + n/2 - 1}{\nu + n/2 - 1} \right)^{\frac{1}{1+\gamma}}}_{=: C_1} \underbrace{\left( \int \bar{\sigma}_\phi(x)^{-\gamma} p_{\mathrm{data}}(x)^{1+\gamma} dx \right)^{\frac{1}{1+\gamma}}}_{P.I.},
\end{aligned} \tag{19}
$$

and the $\gamma$-power cross-entropy is

$$
\begin{aligned}
\mathcal{C}_\gamma(q_\phi, p_\theta) &= -\left( \iint q_\phi(x,z) p_\theta(x,z)^\gamma dx dz \right) \left( \iint p_\theta(x,z)^{1+\gamma} dx dz \right)^{-\frac{\gamma}{1+\gamma}} \\
&= -\underbrace{C^{\frac{\gamma}{1+\gamma}} \left( \frac{\nu + m + n/2 - 1}{\nu - 1} \right)^{-\frac{\gamma}{1+\gamma}}}_{=: C_2} \\
&\quad \times \mathbb{E}_{x \sim p_{\mathrm{data}}} \left[ 1 + \frac{1}{\nu} \|\mu_\phi(x)\|_1 + \frac{\nu + n/2}{\nu(\nu + n/2 - 1)} \|r_\phi(x)\|_1 + \frac{1}{\nu\sigma^2} \mathbb{E}_{z \sim q_\phi(\cdot|x)} \|x - \mu_\theta(z)\|^2 \right].
\end{aligned}
$$

To approximate the power-form integral $P.I.$ in (19), we adopt the following proposition from $t^3$VAE (Kim et al., 2024), which provides a first-order approximation of integrals involving $p_{\text{data}}$ with an $O(\gamma^2)$ error rate.

**Proposition B.1.** *Let $\sigma$ be any positive continuous function, $\gamma \in (-1, 0)$, and suppose the values*

$$H_{j,k} := \mathcal{H}_{j,k}(p_{data}) := \int p_{data}(x)^{1+j\gamma} \left|\log p_{data}(x)\right|^k dx$$

*are finite for each $j \in \{0, 1\}$, $k \in \{1, 2\}$. Then for any compact set $\Omega \subseteq \operatorname{supp} p_{data}$,*

$$\left(\int_\Omega \sigma(x)^{-\gamma} p_{data}(x)^{1+\gamma} dx\right)^{\frac{1}{1+\gamma}} - \int_\Omega \sigma(x)^{-\frac{\gamma}{1+\gamma}} p_{data}(x) dx$$

$$= \gamma \int_\Omega p_{data}(x) \log p_{data}(x) dx + O(\gamma^2).$$

*Proof.* See Proposition 5 in Kim et al. (2024). $\qquad\square$

Using this result, the $\gamma$-entropy of the joint variational distribution $q_\phi(x, z)$ becomes:

$$\mathcal{H}_\gamma(q_\phi) = -C_1 \int \bar{\sigma}_\phi(x)^{-\frac{\gamma}{1+\gamma}} p_{\text{data}}(x) dx + C_1 \mathcal{H}(p_{\text{data}}) + O(\gamma^2),$$

where $\mathcal{H}(p_{\text{data}})$ denotes the differential entropy.

Therefore, the $\gamma$-power divergence $\mathcal{D}_\gamma(q_\phi \,\|\, p_\theta)$ is:

$$D_\gamma(q_\phi \,\|\, p_\theta) = \frac{1}{\gamma} C_\gamma(q_\phi, p_\theta) - \frac{1}{\gamma} H_\gamma(q_\phi)$$

$$\propto \mathbb{E}_{x \sim p_{\text{data}}}\left[C_2\left(\frac{1}{\nu}\|\mu_\phi(x)\|_1 + \frac{\nu + n/2}{\nu(\nu + n/2 - 1)}\|r_\phi(x)\|_1 + \frac{1}{\nu\sigma^2}\mathbb{E}_{z \sim q_\phi(\cdot|x)}\|x - \mu_\theta(z)\|_2^2\right)\right.$$

$$\left. - C_1\bar{\sigma}_\phi(x)^{-\frac{\gamma}{1+\gamma}} + C_1\mathcal{H}(p_{\text{data}})\right] + O(\gamma^2).$$

To make the result simpler, let us define $C_{\text{reg}}$ as the ratio of $C_1$ to $C_2$. By discarding additive constants and the order $\gamma^2$ error, and applying an appropriate rescaling to the $\gamma$-power divergence, we obtain the *raw $\gamma$-loss*:

$$\mathcal{L}_\gamma^{\text{raw}}(\theta, \phi) = \mathbb{E}_{x \sim p_{\text{data}}}\left[\frac{1}{\sigma^2}\mathbb{E}_{z \sim q_\phi(\cdot|x)}\|x - \mu_\theta(z)\|_2^2\right.$$

$$\left. + \underbrace{\|\mu_\phi(x)\|_1 + \frac{\nu + n/2}{(\nu + n/2 - 1)}\|r_\phi(x)\|_1 - \nu C_{\text{reg}}\bar{\sigma}_\phi(x)^{-\frac{\gamma}{1+\gamma}}}_{=:R_\phi \text{ (regularizer term)}}\right]. \quad (20)$$

This loss can be decomposed into an $\ell_2^2$ reconstruction term and a regularization term $R_\phi$, although the latter is less interpretable.

### B.3.2 ALTERNATIVE REPRESENTATION OF THE $\gamma$-LOSS

We now present an alternative form of the $\gamma$-loss, expressed as the sum of a reconstruction term and a $\gamma$-power divergence regularizer. Consider the $\gamma$-power divergence between two symPareto distributions $p_{\text{alt}} = \mathcal{P}_m(0, k\mathbf{1}_m, \nu + n/2)$ and $q_\phi = \mathcal{P}_m(\mu_\phi(x), \sigma_\phi(x), \nu + n/2)$ with $\gamma = -\frac{2}{2\nu + 2m + n}$. Denote

$\nu_n = \nu + n/2$. Under the mild condition $1 + (m+1)\gamma \neq 0$, the divergence can be written as:

$$
\begin{aligned}
\mathcal{D}_\gamma(q_\phi \,\|\, p_{\text{alt}}) &= \frac{1}{\gamma}\mathcal{C}_\gamma(q, p_{\text{alt}}) - \frac{1}{\gamma}\mathcal{H}_\gamma(p_{\text{alt}}) \\
&= C'' \left[ k^{-\frac{m\gamma}{1+\gamma}} \left( 1 + \frac{1}{\nu_n} \left\| \frac{\mu_\phi(x)}{k} \right\|_1 + \frac{1}{\nu_n - 1} \left\| \frac{\sigma_\phi(x)}{k} \left( 1 + \frac{1}{\nu_n} \frac{|\mu_\phi(x)|}{\sigma_\phi(x)} \right)^{-\nu_n + 1} \right\|_1 \right) \right. \\
&\qquad \left. - \bar{\sigma}_\phi(x)^{-\frac{\gamma}{1+\gamma}} \left( \frac{\nu_n + m - 1}{\nu_n - 1} \right) \right] \\
&= C'' k^{-\frac{m\gamma}{1+\gamma}-1} \frac{1}{\nu_n} \left[ \|\mu_\phi(x)\|_1 + \frac{\nu_n}{\nu_n - 1} \left\| \sigma_\phi(x) \left( 1 + \frac{1}{\nu_n} \frac{|\mu_\phi(x)|}{\sigma_\phi(x)} \right)^{-\nu_n + 1} \right\|_1 \right. \\
&\qquad \left. - \bar{\sigma}_\phi(x)^{-\frac{\gamma}{1+\gamma}} (\nu_n) \left( \frac{\nu_n + m - 1}{\nu_n - 1} \right) k^{\frac{m\gamma}{1+\gamma}+1} \right] + \text{const},
\end{aligned}
\tag{21}
$$

where $C'' = (\nu_n + m) C_{m,\nu_n-1,\nu_n}^{\frac{\gamma}{1+\gamma}}$. To align $R_\phi$ with the above $\gamma$-power divergence, we determine $k$ such that

$$
\nu C_{\text{reg}} = (\nu + n/2) \left( \frac{\nu + n/2 + m - 1}{\nu + n/2 - 1} \right) k^{\frac{m\gamma}{1+\gamma}+1}.
$$

Thus, the $\gamma$-loss can be rewritten as the sum of the $\ell_2$ reconstruction term and the $\gamma$-power divergence between encoder and alternative prior. Finally, multiplying by a factor of 2 ensures consistency with the Gaussian MSE scaling, yielding the alternative $\gamma$-loss:

$$
\mathcal{L}_\gamma^{\text{alt}}(\theta, \phi) = \mathbb{E}_{x \sim p_{\text{data}}} \left[ \frac{1}{2\sigma^2} \mathbb{E}_{z \sim q_\phi(\cdot|x)} \|x - \mu_\theta(z)\|_2^2 + \alpha \mathcal{D}_\gamma(q_\phi \,\|\, p_{\text{alt}}) \right].
\tag{22}
$$

The simplified forms of $k$ and $\alpha$ are given as follows, with derivations provided below.

$$
k = \frac{2}{2 + n/\nu} \left( \sigma^{-n} \pi^{-n/2} C_{n/2, \nu-1, \nu} \right)^{1/(\nu+n/2-1)}, \qquad \alpha = -\frac{\gamma \nu}{2 C_2} > 0.
$$

**Derivation of $k$ and $\alpha$ (ParetoVAE with $t$ decoder)**   First, we rewrite the ratio

$$
\begin{aligned}
\frac{C_{m,\nu+n/2,\nu+n/2}}{C} &= \frac{C_{m,\nu+n/2,\nu+n/2}}{C_{m,\nu,\nu} T_{n,2\nu+2m} \sigma^{-n} \left( 2 + \frac{2m}{\nu} \right)^{\frac{n}{2}}} \\
&= \sigma^n (\pi \nu)^{\frac{n}{2}} \frac{\Gamma(\nu)}{\Gamma(\nu + \frac{n}{2})} \left( \frac{\nu}{\nu + \frac{n}{2}} \right)^m \\
&= \sigma^n \pi^{\frac{n}{2}} C_{n/2,\nu,\nu}^{-1} \left( \frac{\nu}{\nu + \frac{n}{2}} \right)^m.
\end{aligned}
$$

Using this identity, $k^{\frac{m\gamma}{1+\gamma}+1}$ reduces to

$$
\begin{aligned}
&\left( \frac{\nu}{\nu + n/2} \right) \left( \frac{\nu + n/2 - 1}{\nu + n/2 + m - 1} \right) C_{\text{reg}} \\
&= \left( \frac{\nu}{\nu + n/2} \right) \frac{C_{m,\nu+n/2,\nu+n/2}^{\frac{\gamma}{1+\gamma}}}{C^{\frac{\gamma}{1+\gamma}}} \left( \frac{\nu + n/2 + m - 1}{\nu + n/2 - 1} \right)^{\frac{\gamma}{1+\gamma}} \left( \frac{\nu + m + n/2 - 1}{\nu - 1} \right)^{\frac{\gamma}{1+\gamma}} \\
&= \left( \frac{\nu}{\nu + n/2} \right) \left( \frac{\nu + n/2 - 1}{\nu - 1} \right)^{\frac{\gamma}{1+\gamma}} \frac{C_{m,\nu+n/2,\nu+n/2}^{\frac{\gamma}{1+\gamma}}}{C^{\frac{\gamma}{1+\gamma}}} \\
&= \left( \frac{\nu}{\nu + n/2} \right)^{1+\frac{m\gamma}{1+\gamma}} \left\{ \left( \frac{\nu + n/2 - 1}{\nu - 1} \right) \sigma^n \pi^{\frac{n}{2}} C_{n/2,\nu,\nu}^{-1} \right\}^{\frac{\gamma}{(1+\gamma)}} \\
&= \left( \frac{\nu}{\nu + n/2} \right)^{1+\frac{m\gamma}{1+\gamma}} \left\{ \sigma^n \pi^{\frac{n}{2}} C_{n/2,\nu-1,\nu}^{-1} \right\}^{\frac{\gamma}{(1+\gamma)}}.
\end{aligned}
$$

Since

$$\frac{\gamma}{1+\gamma} = -\frac{2}{2\nu + 2m + n - 2}, \quad \frac{m\gamma}{1+\gamma} + 1 = \frac{-2m}{2\nu + 2m + n - 2} + 1 = \frac{2\nu + n - 2}{2\nu + 2m + n - 2},$$

we simplify the constant $k$ as follows:

$$k = \left[ \left( \frac{\nu}{\nu + n/2} \right) \left( \frac{\nu + n/2 - 1}{\nu + n/2 + m - 1} \right) C_{\text{reg}} \right]^{\frac{1+\gamma}{1+(m+1)\gamma}}$$

$$= \frac{2}{2 + \nu^{-1}n} \left( \sigma^{-n} \pi^{-\frac{n}{2}} C_{n/2, \nu-1, \nu} \right)^{\frac{1}{\nu + n/2 - 1}}$$

Finally, we simplify the constant $\alpha$ by using the above result:

$$\alpha = \frac{1}{2} C''^{-1} k^{\frac{m\gamma}{1+\gamma} + 1} (\nu + n/2)$$

$$= \frac{1}{2} (\nu + n/2 + m)^{-1} C_{m, \nu+n/2-1, \nu+n/2}^{-\frac{\gamma}{1+\gamma}} \frac{\nu(\nu + n/2 - 1)}{\nu + n/2 + m - 1} \frac{C_1}{C_2}$$

$$= \frac{\nu}{2\nu + 2m + n} \frac{1}{C_2}$$

$$= -\frac{\gamma\nu}{2C_2}.$$

### B.4   PROOF OF PROPOSITION 3.2

*Proof.* We begin by deriving the limits of the normalizing constants $C_{n,\nu,\nu}$ and $T_{n,\nu}$ as $\nu \to \infty$. In (16). we already show that

$$\lim_{\nu \to \infty} C_{n,\nu,\nu} = \frac{1}{2^n}$$

by using the Stirling's approximation. Using the same approximation, we obtain

$$\lim_{\nu \to \infty} T_{n,\nu} = \pi^{-\frac{n}{2}} \lim_{\nu \to \infty} \frac{\Gamma(\frac{\nu+n}{2})}{\nu^{\frac{n}{2}} \Gamma(\frac{\nu}{2})}$$

$$= \pi^{-\frac{n}{2}} \lim_{\nu \to \infty} \sqrt{\frac{\nu + n - 2}{\nu - 2}} \frac{\left(1 + \frac{n}{\nu - 2}\right)^{\frac{\nu-2}{2}} \left(\frac{\nu+n-2}{2e}\right)^{\frac{n}{2}}}{\nu^{\frac{n}{2}}}$$

$$= (2\pi)^{-\frac{n}{2}}.$$

Since $\frac{\gamma}{1+\gamma} \to 0$ as $\nu \to \infty$ and the base inside the exponent remains bounded away from zero and infinity, it follows that

$$\lim_{\nu \to \infty} C_2 = \lim_{\nu \to \infty} \left\{ C_{m,\nu,\nu} T_{n, 2\nu+2m} \sigma^{-n} \left( 2 + \frac{2m}{\nu} \right)^{\frac{n}{2}} \left( \frac{\nu - 1}{\nu + m + n/2 - 1} \right) \right\}^{\frac{\gamma}{1+\gamma}} = 1$$

and thus

$$\lim_{\nu \to \infty} \alpha = \lim_{\nu \to \infty} \frac{2\nu}{2\nu + 2m + n} \cdot \frac{1}{2C_2} = \frac{1}{2}. \tag{23}$$

Next, we derive the limit of $k$ and $\beta$, which follows directly from the expression.

$$\lim_{\nu \to \infty} k = \lim_{\nu \to \infty} \frac{2}{2 + \nu^{-1}n} \left( \sigma^{-n} \pi^{-\frac{n}{2}} C_{n/2, \nu-1, \nu} \right)^{\frac{1}{\nu + n/2 - 1}} = 1.$$

$$\lim_{\nu \to \infty} \beta = \lim_{\nu \to \infty} \left( 1 + \frac{n}{\nu} \right) C_{n, \nu-1, \nu}^{\frac{\gamma}{1+\gamma}} \bar{\sigma}_2^{-\frac{\gamma}{1+\gamma}}$$

$$= 1.$$

Next, we compute the limit of $\mathcal{D}_\gamma(q_\phi \| p_{\text{alt}})$ in (23). As $\nu \to \infty$, each term converges to:

$$\lim_{\nu \to \infty} C'' \left[ k^{-\frac{m\gamma}{1+\gamma}} - \bar\sigma_\phi(x)^{-\frac{m\gamma}{1+\gamma}} \left( \frac{\nu + n/2 + m - 1}{\nu + n/2 - 1} \right) \right] = m \log \bar\sigma_\phi(x) - m$$

$$\lim_{\nu \to \infty} \frac{C''}{\nu + \frac{n}{2}} \left\| \frac{\mu_\phi(x)}{k} \right\|_1 = \|\mu_\phi(x)\|_1$$

$$\lim_{\nu \to \infty} \frac{C''}{\nu + n/2 - 1} \left\| \frac{\sigma_\phi(x)}{k} \left( 1 + \frac{1}{\nu + n/2} \frac{|\mu_\phi(x)|}{\sigma_\phi(x)} \right)^{-(\nu + n/2) + 1} \right\|_1 = \left\| \sigma_\phi(x) \exp\left( -\frac{|\mu_\phi(x)|}{\sigma_\phi(x)} \right) \right\|_1.$$

Combining these limits, we obtain

$$\lim_{\nu \to \infty} \mathcal{D}_\gamma(q_\phi \| p_{\text{alt}}) = \sum_{i=1}^m \left[ \sigma_{\phi,i}(x) \exp\left( -\frac{|\mu_{\phi,i}(x)|}{\sigma_{\phi,i}(x)} \right) + |\mu_{\phi,i}(x)| + \log \sigma_{\phi,i}(x) - 1 \right].$$

To verify that this limiting form coincides with the KL divergence between two product form of univariate Laplace, consider $q(z) = \prod_{i=1}^m q_i(z_i)$ and $p(z) = \prod_{i=1}^m p_i(z_i)$, where $q_i = L_1(z_i \mid \mu_{1,i}, b_{1,i})$ and $p_i = L_1(z_i \mid \mu_{2,i}, b_{2,i})$.

The KL divergence between $p(z)$ and $q(z)$ can be factorized as

$$\mathcal{D}_{\text{KL}}(q \| p) = \sum_{i=1}^m \mathcal{D}_{\text{KL}}(q_i \| p_i)$$

$$= \sum_{i=1}^m \frac{b_{1,i} \exp\left( -\frac{|\mu_{1,i} - \mu_{2,i}|}{b_{1,i}} \right) + |\mu_{1,i} - \mu_{2,i}|}{b_{2,i}} + \log\left( \frac{b_{2,i}}{b_{1,i}} \right) - 1.$$

By setting $\mu_{1,i} = \mu_{\phi,i}(x)$, $\mu_{2,i} = 0$, and $b_{1,i} = \sigma_{\phi,i}(x)$, the distributions $p(z)$ and $q(z)$ correspond to the limiting forms of $q_\phi$ and $p_{\text{alt}}(z)$ in Theorem A.1. We therefore conclude that

$$\lim_{\nu \to \infty} \mathcal{D}_\gamma(q_\phi \| p_{\text{alt}}) = \mathcal{D}_{\text{KL}}(q_{\phi,\infty} \| p_{\text{alt},\infty}).$$

$\square$

## B.5 DERIVATION OF THE HEAVY-TAILED VAEs WITH SYMPARETO DECODER

### B.5.1 PARETOVAE WITH SYMPARETO DECODER

To match the exponent in (11), we set $\gamma = -\frac{1}{\nu + m + n}$. Since the joint decoder follows a symPareto structure, its normalizing constant is $C = C_{n+m,\nu,\nu}\, \sigma^{-n}$. We then compute the $\gamma$-entropy and $\gamma$-cross entropy for the joint distributions $p_\theta$ and $q_\phi$.

The only difference from Appendix B.3 lies in the terms involving $p_\theta(x|z)$. Repeating the same steps as Appendix B.3, we obtain

$$\iint p_\theta(x,z)^{1+\gamma} dx dz = \mathbb{E}_{z \sim p_Z} \mathbb{E}_{x \sim p_\theta(\cdot|z)} \left[ p_\theta(x,z)^\gamma \right]$$

$$= C^\gamma \mathbb{E}_{z \sim p_Z} \mathbb{E}_{x \sim p_\theta(\cdot|z)} \left[ 1 + \frac{1}{\nu} \left( \|z\|_1 + \frac{1}{\sigma} \|x - \mu_\theta(z)\|_1 \right) \right]$$

$$= C^\gamma \mathbb{E}_{z \sim p_Z} \left[ 1 + \frac{1}{\nu} \|z\|_1 + \frac{n}{\nu} \frac{\nu + m}{\nu + m - 1} \cdot \frac{(\nu + \|z\|_1)}{\nu + m} \right]$$

$$= C^\gamma \mathbb{E}_{z \sim p_Z} \left[ \left( 1 + \frac{n}{\nu + m - 1} \right) \left( 1 + \frac{1}{\nu} \|z\|_1 \right) \right]$$

$$= C^\gamma \left( 1 + \frac{n}{\nu + m - 1} \right) \left( 1 + \frac{m}{\nu - 1} \right)$$

$$= C^\gamma \left( \frac{\nu + n + m - 1}{\nu - 1} \right),$$

$$\iint q_\phi(x,z)^{1+\gamma}dxdz = \int \left( \int q_\phi(z|x)^{1+\gamma}dz \right) p_{\text{data}}(z)^{1+\gamma}dx$$

$$= C_{m,\nu+n,\nu+n}^{1+\gamma}C_{m,\nu+n-1,\nu+n}^{-1} \int |\sigma_\phi(x)|^{-\gamma}p_{\text{data}}(x)^{1+\gamma}dx,$$

$$\iint q_\phi(x,z)p_\theta(x,z)^\gamma dxdz = \mathbb{E}_{x\sim p_{\text{data}}}\mathbb{E}_{z\sim q_\phi(\cdot|x)}\left[p_\theta(x,z)^\gamma\right]$$

$$= C^\gamma \mathbb{E}_{x\sim p_{\text{data}}}\mathbb{E}_{z\sim q_\phi(\cdot|x)}\left[1 + \frac{1}{\nu}\left(\|z\|_1 + \frac{1}{\sigma}\|x - \mu_\theta(z)\|_1\right)\right]$$

$$= C^\gamma \mathbb{E}_{x\sim p_{\text{data}}}\left[1 + \frac{1}{\nu}\|\mu_\phi(x)\|_1 + \frac{\nu+n}{\nu(\nu+n-1)}\|\tilde{r}_\phi(x)\|_1 + \frac{1}{\nu\sigma}\mathbb{E}_{z\sim q_\phi(\cdot|x)}\|x - \mu_\theta(z)\|_1\right],$$

where

$$\tilde{r}_\phi(x) := \sigma_\phi(x)\left(1 + \frac{|\mu_\phi(x)|}{(\nu+n)\sigma_\phi(x)}\right)^{-(\nu+n-1)}.$$

Hence, the $\gamma$-cross entropy is given by:

$$\mathcal{C}_\gamma(q_\phi, p_\theta) = -\left(\iint q_\phi(x,z)p_\theta(x,z)^\gamma dxdz\right)\left(\iint p_\theta(x,z)^{1+\gamma}dxdz\right)^{-\frac{\gamma}{1+\gamma}}$$

$$= -\underbrace{C^{\frac{\gamma}{1+\gamma}}\left(\frac{\nu+n+m-1}{\nu-1}\right)^{-\frac{\gamma}{1+\gamma}}}_{=:C_2}$$

$$\times \mathbb{E}_{x\sim p_{\text{data}}}\left[1 + \frac{1}{\nu}\|\mu_\phi(x)\|_1 + \frac{\nu+n}{\nu(\nu+n-1)}\|\tilde{r}_\phi(x)\|_1 + \frac{1}{\nu\sigma}\mathbb{E}_{z\sim q_\phi(\cdot|x)}\|x - \mu_\theta(z)\|_1\right],$$

and the $\gamma$-power divergence $\mathcal{D}_\gamma(q_\phi \,\|\, p_\theta)$ becomes:

$$D_\gamma(q_\phi \,\|\, p_\theta) = \frac{1}{\gamma}C_\gamma(q_\phi, p_\theta) - \frac{1}{\gamma}H_\gamma(q_\phi)$$

$$\propto \mathbb{E}_{x\sim p_{\text{data}}}\left[C_2\left(\frac{1}{\nu\sigma}\mathbb{E}_{z\sim q_\phi(\cdot|x)}\|x - \mu_\theta(z)\|_1 + \frac{1}{\nu}\|\mu_\phi(x)\|_1 + \frac{\nu+n}{\nu(\nu+n-1)}\|\tilde{r}_\phi(x)\|_1\right)\right.$$

$$\left. -C_1\bar{\sigma}_\phi(x)^{-\frac{\gamma}{1+\gamma}} + C_1\mathcal{H}(p_{\text{data}})\right] + O(\gamma^2)$$

where $C_1$ is as defined in (19), and the error term $O(\gamma^2)$ follows from the application of Proposition B.1.

Discarding additive constants and up to anirrelevant positive scaling factor (cf. Appendix B.3) and use $C_{\text{reg}} := C_1/C_2$ again, the *raw* $\gamma$-loss can be written as

$$\mathcal{L}_\gamma(\theta, \phi) = \mathbb{E}_{x\sim p_{\text{data}}}\left[\frac{1}{\sigma}\mathbb{E}_{z\sim q_\phi(\cdot|x)}\|x - \mu_\theta(z)\|_1\right. \tag{24}$$

$$\left. + \underbrace{\|\mu_\phi(x)\|_1 + \frac{\nu+n}{(\nu+n-1)}\|\tilde{r}_\phi(x)\|_1 - \nu C_{\text{reg}}\bar{\sigma}_\phi(x)^{-\frac{\gamma}{1+\gamma}}}_{=:R_\phi(\text{regularizer})}\right]. \tag{25}$$

Compared with (20), the only difference lies in the reconstruction term, changing square of $\ell_2$ norm to $\ell_1$ one.

We further represent alternative form, by solving the equation with respect to $k$:

$$\nu C_{\text{reg}} = (\nu+n)\left(\frac{\nu+n+m-1}{\nu+n-1}\right)k^{\frac{m\gamma}{1+\gamma}+1},$$

Then, the alternative form of the $\gamma$-loss becomes

$$\mathcal{L}_\gamma^{\text{alt}}(\theta, \phi) = \mathbb{E}_{x\sim p_{\text{data}}}\left[\frac{1}{\sigma}\mathbb{E}_{z\sim q_\phi(\cdot|x)}\|x - \mu_\theta(z)\|_1 + \alpha D_\gamma(q_\phi \,\|\, p_{\text{alt}})\right],$$

where $p_{\text{alt}} = \mathcal{P}_m(0, k\mathbf{1}_m, \nu+n)$ and $k = \left(\frac{\nu}{\nu+n}\right)(C_{n,\nu-1,\nu}\sigma^{-n})^{\frac{1}{\nu+n-1}}$, $\alpha = -\frac{\gamma\nu}{C_2}$.

**Derivation of $k$ and $\alpha$ (ParetoVAE with symPareto decoder)** We first simplify $C_{\text{reg}}$ to obtain expressions for $k$ and $\alpha$: Since

$$\frac{C_{m,\nu+n,\nu+n}}{C_{n+m,\nu,\nu}} = \frac{\Gamma(\nu+n+m)}{2^m(\nu+n)^m\Gamma(\nu+n)}\frac{2^{m+n}\nu^{m+n}\Gamma(\nu)}{\Gamma(\nu+n+m)}$$

$$= \frac{2^n\nu^n\Gamma(\nu)}{\Gamma(\nu+n)}\left(\frac{\nu}{\nu+n}\right)^m$$

$$= C_{n,\nu,\nu}^{-1}\left(\frac{\nu}{\nu+n}\right)^m,$$

it follows that

$$C_{\text{reg}} := \frac{C_1}{C_2} = \frac{\left(\frac{\nu+n+m-1}{\nu+n-1}\right)^{\frac{1}{1+\gamma}} C_{m,\nu+n,\nu+n}^{\frac{\gamma}{1+\gamma}}}{C^{\frac{\gamma}{1+\gamma}}\left(\frac{\nu+n+m-1}{\nu-1}\right)^{-\frac{\gamma}{1+\gamma}}}$$

$$= \frac{\nu+n+m-1}{(\nu-1)^{\frac{\gamma}{1+\gamma}}(\nu+n-1)^{\frac{1}{1+\gamma}}}\left(\frac{C_{m,\nu+n,\nu+n}}{C_{n+m,\nu,\nu}\sigma^{-n}}\right)^{\frac{\gamma}{1+\gamma}}$$

$$= \frac{\nu+n+m-1}{(\nu-1)^{\frac{\gamma}{1+\gamma}}(\nu+n-1)^{\frac{1}{1+\gamma}}}C_{n,\nu,\nu}^{-\frac{\gamma}{1+\gamma}}\left(\frac{\nu}{\nu+m}\right)^{\frac{m\gamma}{1+\gamma}}\sigma^{\frac{n\gamma}{1+\gamma}}.$$

This yields that

$$k = \left(\frac{\nu}{\nu+n}\cdot\frac{\nu+n-1}{\nu+n+m-1}C_{\text{reg}}\right)^{\frac{1+\gamma}{1+(m+1)\gamma}}$$

$$= \left\{\left(\frac{\nu}{\nu+n}\right)^{\frac{m\gamma}{1+\gamma}+1}\left(\frac{\nu+n-1}{\nu-1}\right)^{\frac{\gamma}{1+\gamma}}C_{n,\nu,\nu}^{-\frac{\gamma}{1+\gamma}}\sigma^{\frac{n\gamma}{1+\gamma}}\right\}^{\frac{1+\gamma}{1+(m+1)\gamma}}$$

$$= \left(\frac{\nu}{\nu+n}\right)\left\{\left(\frac{\nu+n-1}{\nu-1}\right)C_{n,\nu-1,\nu}^{-1}\sigma^n\right\}^{\frac{\gamma}{1+(m+1)\gamma}}$$

$$= \left(\frac{\nu}{\nu+n}\right)\left(C_{n,\nu-1,\nu}\sigma^{-n}\right)^{\frac{1}{\nu+n-1}}.$$

Finally, $\alpha$ is given by

$$\alpha = \frac{\nu+n}{\nu+n+m}C_{m,\nu+n-1,\nu+n}^{-\frac{\gamma}{1+\gamma}}k^{\frac{m\gamma}{1+\gamma}+1}$$

$$= \frac{\nu+n}{\nu+n+m}C_{m,\nu+n-1,\nu+n}^{-\frac{\gamma}{1+\gamma}}\cdot\frac{\nu}{\nu+n}\cdot\frac{\nu+n-1}{\nu+n+m-1}\frac{C_1}{C_2}$$

$$= \frac{\nu}{\nu+n+m}C_{m,\nu+n-1,\nu+n}^{-1}C_{m,\nu+n,\nu+n}\cdot\frac{\nu+n-1}{\nu+n+m-1}\frac{1}{C_2}$$

$$= -\frac{\gamma\nu}{C_2}.$$

### B.5.2 $t^3$VAE WITH SYMPARETO DECODER

From Appendix B.2, the joint decoder distribution is

$$p_\theta(x,z) \propto \left[1+\frac{1}{\nu}\left(\|z\|_2^2+\frac{1}{\sigma}\|x-\mu_\theta(z)\|_1\right)\right]^{-\left(\frac{\nu+m}{2}+n\right)}.$$

The corresponding prior and decoder are

$$p(z) = t_m(z\mid 0, I_m, \nu), \qquad p_\theta(x\mid z) = \mathcal{P}_n(x\mid\mu_\theta(z), \frac{2(\nu+\|z\|_2^2)}{\nu+m}\sigma\mathbf{1}_n, \frac{\nu+m}{2}).$$

We also use the encoder distribution

$$q_\phi(z \mid x) = t_m\left(z \mid \mu_\phi(x), \sigma_\phi(x)^2 I_m, \nu + 2n\right), \quad \sigma_\phi(x) \in \mathbb{R}_+^m,$$

which differs slightly from Kim et al. (2024) (diagonal covariance, no $(1 + \nu^{-1}n)^{-1}$ scaling). This simplification is harmless, since $\sigma_\phi(x)$ is learnable and typically assumed diagonal in practice.

We first derive the normalizing constant of the joint decoder:

$$C = T_{m,\nu} C_{n, \frac{\nu+m}{2}, \frac{\nu+m}{2}} \sigma^{-n} \left(\frac{\nu + m}{2\nu}\right)^{\frac{n}{2}}.$$

Set $\gamma = -\frac{2}{\nu+m+2n}$ to match the exponent in the joint decoder distribution. We then derive the $\gamma$-entropy and $\gamma$-cross entropy for the joint distributions $p_\theta$ and $q_\phi$. From Kim et al. (2024), the $\gamma$-entropy is

$$\mathcal{H}_\gamma(q_\phi) = - \underbrace{T_{m,\nu+2n}^{\frac{\gamma}{1+\gamma}} \left(\frac{\nu + m + 2n - 2}{\nu + 2n - 2}\right)^{\frac{1}{1+\gamma}}}_{=: C_1} \left(\int \bar{\sigma}_\phi(x)^{-\gamma} p_{\text{data}}(x)^{1+\gamma} dx\right)^{\frac{1}{1+\gamma}}.$$

For the joint distribution $p_\theta$, we compute

$$\iint p_\theta(x,z)^{1+\gamma} dx dz = \mathbb{E}_{z \sim p_Z} \mathbb{E}_{x \sim p_\theta(\cdot|z)} \left[p_\theta(x,z)^\gamma\right]$$

$$= C^\gamma \mathbb{E}_{z \sim p_Z} \mathbb{E}_{x \sim p_\theta(\cdot|z)} \left[1 + \frac{1}{\nu}\left(\|z\|_2^2 + \frac{1}{\sigma}\|x - \mu_\theta(z)\|_1\right)\right]$$

$$= C^\gamma \mathbb{E}_{z \sim p_Z} \left[1 + \frac{1}{\nu}\|z\|_2^2 + \frac{n}{\nu}\frac{\nu+m}{\nu+m-2} \cdot \frac{2(\nu + \|z\|_2^2)}{\nu+m}\right]$$

$$= C^\gamma \mathbb{E}_{z \sim p_Z} \left[\left(1 + \frac{2n}{\nu+m-2}\right)\left(1 + \frac{1}{\nu}\|z\|_2^2\right)\right]$$

$$= C^\gamma \left(1 + \frac{2n}{\nu+m-2}\right)\left(1 + \frac{m}{\nu-2}\right)$$

$$= C^\gamma \left(\frac{\nu + 2n + m - 2}{\nu - 2}\right).$$

$$\iint q_\phi(x,z) p_\theta(x,z)^\gamma dx dz = \mathbb{E}_{x \sim p_{\text{data}}} \mathbb{E}_{z \sim q_\phi(\cdot|x)} \left[p_\theta(x,z)^\gamma\right]$$

$$= C^\gamma \mathbb{E}_{x \sim p_{\text{data}}} \mathbb{E}_{z \sim q_\phi(\cdot|x)} \left[1 + \frac{1}{\nu}\left(\|z\|_2^2 + \frac{1}{\sigma}\|x - \mu_\theta(z)\|_1\right)\right]$$

$$= C^\gamma \mathbb{E}_{x \sim p_{\text{data}}} \left[1 + \frac{1}{\nu}\|\mu_\phi(x)\|_2^2 + \frac{\nu+2n}{\nu(\nu+2n-2)}\|\sigma_\phi(x)\|_1 + \frac{1}{\nu\sigma}\mathbb{E}_{z \sim q_\phi(\cdot|x)}\|x - \mu_\theta(z)\|_1\right].$$

Thus,

$$\mathcal{C}_\gamma(q_\phi, p_\theta) = -\left(\iint q_\phi(x,z) p_\theta(x,z)^\gamma dx dz\right) \left(\iint p_\theta(x,z)^{1+\gamma} dx dz\right)^{-\frac{\gamma}{1+\gamma}}$$

$$= - \underbrace{C^{\frac{\gamma}{1+\gamma}} \left(\frac{\nu + 2n + m - 2}{\nu - 2}\right)^{-\frac{\gamma}{1+\gamma}}}_{=: C_2}$$

$$\times \mathbb{E}_{x \sim p_{\text{data}}} \left[1 + \frac{1}{\nu}\|\mu_\phi(x)\|_2^2 + \frac{\nu+2n}{\nu(\nu+2n-2)}\|\sigma_\phi(x)\|_1 + \frac{1}{\nu\sigma}\mathbb{E}_{z \sim q_\phi(\cdot|x)}\|x - \mu_\theta(z)\|_1\right].$$

and with Proposition B.1, the $\gamma$-power divergence $\mathcal{D}_\gamma(q_\phi \,\|\, p_\theta)$ becomes:

$$
\begin{aligned}
D_\gamma(q_\phi \,\|\, p_\theta) &= \frac{1}{\gamma} C_\gamma(q_\phi, p_\theta) - \frac{1}{\gamma} H_\gamma(q_\phi) \\
&\propto \mathbb{E}_{x \sim p_{\text{data}}}\Bigg[ C_2 \left( \frac{1}{\nu\sigma} \mathbb{E}_{z \sim q_\phi(\cdot|x)} \|x - \mu_\theta(z)\|_1 + \frac{1}{\nu} \|\mu_\phi(x)\|_2^2 + \frac{\nu + 2n}{\nu(\nu + 2n - 2)} \|\sigma_\phi(x)\|_1 \right) \\
&\qquad\qquad - C_1 \bar{\sigma}_\phi(x)^{-\frac{\gamma}{1+\gamma}} + C_1 \mathcal{H}(p_{\text{data}}) \Bigg] + O(\gamma^2).
\end{aligned}
$$

Following the same steps as before, the raw $\gamma$-loss is

$$
\begin{aligned}
\mathcal{L}_\gamma^{\text{raw}}(\theta, \phi) = \mathbb{E}_{x \sim p_{\text{data}}}\Bigg[ &\frac{1}{\sigma} \mathbb{E}_{z \sim q_\phi(\cdot|x)} \|x - \mu_\theta(z)\|_1 \\
&+ \underbrace{\|\mu_\phi(x)\|_2^2 + \frac{\nu + 2n}{(\nu + 2n - 2)} \|\sigma_\phi(x)\|_1 - \nu C_{\text{reg}} \bar{\sigma}_\phi(x)^{-\frac{\gamma}{1+\gamma}}}_{=:R_\phi \text{(regularizer)}} \Bigg].
\end{aligned}
$$

Compared with the previous alternative $\gamma$-loss of ParetoVAE models (22 and 24), it is notable that the $\ell_1$ regularization term of $\mu_\phi(x)$ is changed to the $\ell_2^2$ term and $\|r_\phi(x)\|_1$ (or $\|\tilde{r}_\phi(x)\|_1$) is simplified to $\|\sigma_\phi(x)\|_1$.

We finally obtain an alternative representation of the raw $\gamma$-loss by solving the equation with respect to $k$:

$$
\nu C_{\text{reg}} = (\nu + 2n) \left( \frac{\nu + 2n + m - 2}{\nu + 2n - 2} \right) k^{\frac{m\gamma}{1+\gamma} + 2},
$$

Accordingly, the $\gamma$-loss can be rewritten as

$$
\mathcal{L}_\gamma^{\text{alt}}(\theta, \phi) = \mathbb{E}_{x \sim p_{\text{data}}}\left[ \frac{1}{\sigma} \mathbb{E}_{z \sim q_\phi(\cdot|x)} \|x - \mu_\theta(z)\|_1 + \alpha D_\gamma(q_\phi \,\|\, p_{\text{alt}}) \right],
$$

where $p_{\text{alt}} = t_m(0, k^2 I_m, \nu + 2n)$, $\alpha = -\frac{\gamma\nu}{2C_2}$, and

$$
k = \left( \frac{\nu}{\nu + 2n} \right)^{\frac{1}{2}} \left[ (2\sigma^2)^{-\frac{n}{2}} C_{n, \frac{\nu}{2}, \frac{\nu}{2}} \left( \frac{\nu}{\nu + m} \right)^{\frac{n}{2}} \right]^{\frac{2}{\nu + 2n - 1}}.
$$

**Derivation of $k$ and $\alpha$ ($t^3$VAE with Pareto decoder)** We first compute $C_{\text{reg}}$ as:

$$
\begin{aligned}
C_{\text{reg}} := \frac{C_1}{C_2} &= \frac{T_{m,\nu+2n}^{\frac{\gamma}{1+\gamma}} \left( \frac{\nu + m + 2n - 2}{\nu + 2n - 2} \right)^{\frac{1}{1+\gamma}}}{\left( T_{m,\nu} C_{n, \frac{\nu+m}{2}, \frac{\nu+m}{2}} \sigma^{-n} \left( \frac{1}{2} + \frac{m}{2\nu} \right)^{\frac{n}{2}} \frac{\nu - 2}{\nu + 2n + m - 2} \right)^{\frac{\gamma}{1+\gamma}}} \\
&= \left( \frac{\nu + m + 2n - 2}{\nu + 2n - 2} \right) \left( \nu^{-\frac{n}{2}} T_{m,\nu+2n}^{-1} T_{m,\nu} C_{n, \frac{\nu+m}{2}, \frac{\nu+m}{2}} \sigma^{-n} \left( \frac{\nu + m}{2} \right)^{\frac{n}{2}} \right)^{-\frac{\gamma}{1+\gamma}} \\
&= \left( \frac{\nu + m + 2n - 2}{\nu + 2n - 2} \right) \left( C_{n, \frac{\nu}{2}, \frac{\nu}{2}} \left( 1 + \frac{2n}{\nu} \right)^{\frac{m}{2}} \left( \frac{\nu}{\nu + m} \right)^{\frac{n}{2}} (2\sigma^2)^{-\frac{n}{2}} \right)^{-\frac{\gamma}{1+\gamma}}
\end{aligned}
$$

This yields a simpler form of $k$ as

$$
\begin{aligned}
k &= \left( \frac{\nu}{\nu + 2n} \cdot \frac{\nu + 2n - 2}{\nu + n + 2m - 2} C_{\text{reg}} \right)^{\frac{1+\gamma}{2 + (m+1)\gamma}} \\
&= \left( \frac{\nu}{\nu + 2n} \right)^{\frac{1}{2}} \left[ (2\sigma^2)^{-\frac{n}{2}} C_{n, \frac{\nu}{2}, \frac{\nu}{2}} \left( \frac{\nu}{\nu + m} \right)^{\frac{n}{2}} \right]^{\frac{2}{\nu + 2n - 1}}.
\end{aligned}
$$

Finally, $\alpha$ is given by

$$
\begin{aligned}
\alpha &= \frac{\nu + 2n}{C''} k^{\frac{m\gamma}{1+\gamma}+2} \\
&= -\gamma(\nu + 2n) \left( \frac{\nu + 2n - 2}{\nu + 2n + m - 2} \right)^{-\frac{\gamma}{1+\gamma}} T_{m,\nu+2n}^{-\frac{\gamma}{1+\gamma}} \frac{\nu}{\nu + 2n} \frac{\nu + 2n - 2}{\nu + 2n + m - 2} \frac{C_1}{C_2} \\
&= -\frac{\gamma\nu}{C_2}.
\end{aligned}
$$

## C EXPERIMENTAL DETAILS

**Common Environment**   All experiments are conducted using Python 3.12 and PyTorch 2.8 on a Linux Ubuntu 24.04 LTS system equipped with Intel® Xeon® Platinum 8568Y+ @ 2.3GHz (48 cores) processors and an NVIDIA RTX 6000 Ada GPU (48GB), with CUDA 13.0 and cuDNN 8.9.7. Additional details for each experimental setting are provided below.

### C.1 DETAILS ON GRAPH DEGREE RECONSTRUCTION

The SNAP Epinions social network (Richardson et al., 2003) consists of 508,837 directed edges and 75,879 nodes. From this dataset, we compute the in-degree (the number of incoming edges) and out-degree (the number of outgoing edges) for each node. This results in a two-dimensional dataset with 75,879 (in-degree, out-degree) pairs, which exhibit a power-law distribution.

The encoder and decoder are implemented as 2-layer MLPs with 256 units per layer. The latent dimension is set to 4, with a batch size of 512. We use the LeakyReLU activation function (negative slope 1e-4), followed by batch normalization. The model is trained with the Adam optimizer (Kingma & Ba, 2017), using a learning rate of 0.0001. The dataset are randomly split into train/validation/test sets with proportions of (0.6, 0.2, 0.2).

### C.2 DETAILS ON WORD FREQUENCY ANALYSIS

We utilize the WikiText-2-raw dataset (Merity et al., 2017). All non-English characters are also filtered out. Although the full vocabulary contains 65,653 tokens, we choose to exclude words with a frequency of 4 or less. This is because these rare words, totaling 44,791, constitute approximately 70% of the full vocabulary and serve as a significant impediment to stable model training. The remaining vocabulary of 19,962 tokens retains a sufficiently heavy-tailed distribution, which is characteristic of natural language.

For the analysis, the vocabulary are represented as 19,962-dimensional bag-of-words vectors. The head-and-tail split is defined as the 2,241 most frequent words (head) and the 2,241 least frequent words (tail), corresponding to words appearing exactly five times.

The encoder and decoder are symmetrically structured, each consisting of a 3-layer MLP with 256 units per layer. The latent dimension is set to 64, and the batch size was 512. The ReLU activation function is used for all layers, and batch normalization is applied without the use of dropout. All models are trained using the AdamW optimizer (Loshchilov & Hutter, 2019) with a fixed learning rate of 0.0005, without learning rate scheduling. The regularization weight was linearly scheduled.

### C.3 DETAILS ON IMAGE DENOISING APPLICATION

Salt-and-pepper noise was applied to the datasets, with a noise probability defined as the likelihood of each pixel being corrupted. For corrupted pixels, a salt ratio of 0.5 was used, indicating an equal probability of a pixel being set to its maximum (salt) or minimum (pepper) intensity value.

The MNIST, SVHN, and CIFAR10 datasets are resized to $32 \times 32$ pixels, while the Omniglot and CelebA datasets are resized to 64×64 pixels. The latent dimensions are set to 16 for MNIST, 32 for SVHN, 64 for both CIFAR10 and Omniglot, and 128 for CelebA. A batch size of 512 is used for all datasets.

Table 4: Relative runtime per training step (forward+backward+update) compared to the proposed $\gamma$-loss (normalized to 1.0) on MNIST with batch size 128.

| Method | Latent dim $m = 32$ | Latent dim $m = 64$ |
|---|---|---|
| rule ParetoVAE ($\gamma$-loss) | 1.00 | 1.00 |
| ParetoVAE (ELBO, $K = 1$) | 1.02 | 1.04 |
| ParetoVAE (ELBO, $K = 4$) | 1.06 | 1.33 |
| ParetoVAE (ELBO, $K = 16$) | 1.24 | 1.40 |

The encoder and decoder are constructed symmetrically in all experiments, with the number of stacked residual blocks being the architectural difference between datasets. Specifically, the networks for MNIST, SVHN, and CIFAR10 consist of 6 residual blocks, while the larger networks for Omniglot and CelebA utilized 8 residual blocks.

The models were trained using the AdamW optimizer with a fixed learning rate of 0.0005. No learning rate scheduler is utilized. The regularization weight is selected from three schedules—constant, linear, and logistic—tuned specifically for each model and dataset. We observe that the logistic schedule is more effective for 1-channel images, while the linear schedule yields better results for 3-channel images.

The external classifiers are trained for each of the MNIST, SVHN, CIFAR10, and Omniglot. Their classification accuracies are 0.9976, 0.9611, 0.8452, and 0.8497, respectively. Specifically for Omniglot, the 1,623 character classes are re-categorized to 50 language classes.

### C.4 COMPARISON OF COMPUTATIONAL COST VERSUS ELBO

This section quantifies the computational trade-off between ELBO-based training and our proposed $\gamma$-loss for learning the ParetoVAE. We focus on the setting used in our experiments: a symPareto encoder/prior with a Student's $t$ decoder.

**ELBO decomposition.** For a datapoint $x$, the ELBO can be decomposed as

$$\text{ELBO}(x) = \mathbb{E}_{q_\phi(z|x)}[\log p_\theta(x|z)] - \mathbb{E}_{q_\phi(z|x)}[\log q_\phi(z|x)] + \mathbb{E}_{q_\phi(z|x)}[\log p(z)].$$

With a $t$-decoder, the reconstruction term is of the form

$$\mathbb{E}_{z \sim q_\phi(\cdot|x)}[\log p_\theta(x|z)] \propto \mathbb{E}_{z \sim q_\phi(\cdot|x)}\left[\log\left(1 + \frac{\|x - \mu_\theta(z)\|_2^2}{(\nu + m)\sigma^2}\right)\right].$$

The entropy term $-\mathbb{E}_{q_\phi(z|x)}[\log q_\phi(z|x)]$ admits a closed form for our symPareto encoder; in our notation,

$$\mathbb{E}_{q_\phi(z|x)}[\log q_\phi(z|x)] = -\frac{2(\nu + n + m)}{(\nu + n - 1)^2}C_{n,\nu+n,\nu+n} - \sum_{i=1}^{m}\log\sigma_{\phi,i}(x) + \text{const.}$$

In contrast, the prior cross-entropy term $\mathbb{E}_{q_\phi(z|x)}[\log p(z)]$ has no closed-form expression under the symPareto prior. Consequently, ELBO training requires a Monte Carlo approximation,

$$\mathbb{E}_{q_\phi(z|x)}[\log p(z)] \approx \frac{1}{K}\sum_{j=1}^{K}\log p(z^{(j)}), \qquad z^{(j)} \sim q_\phi(\cdot|x),$$

which (i) increases the wall-clock cost roughly linearly in $K$ and (ii) introduces additional estimator variance. The latter effect is amplified when $q_\phi(z|x)$ is heavy-tailed.

**Wall-clock time comparison** We measured the wall-clock time per training step (forward + backward + optimizer update) on MNIST with batch size 128, and report relative runtime normalized so that the $\gamma$-loss runtime equals 1.0. For reliable timing on GPU, we performed a warm-up phase and synchronized CUDA before and after timing.

Table 4 summarizes results for latent dimensions $m \in \{32, 64\}$ and Monte Carlo sizes $K \in \{1, 4, 16\}$. At $K = 1$, the runtime difference is small. However, the gap grows as either the Monte Carlo size

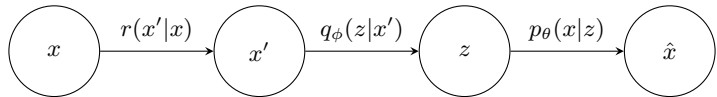

Figure 5: Graphical model of the denoising task

$K$ or the latent dimension $m$ increases. This is consistent with the fact that ELBO training must repeatedly evaluate $\log p(z^{(j)})$ for heavy-tailed samples, increasing both compute and estimator variance. Overall, these measurements support the practical advantage of the $\gamma$-loss formulation when one seeks stable training without increasing Monte Carlo budget.

### C.5 THEORETICAL JUSTIFICATION OF DENOISING WITH PARETOVAE

In this section, we formalize the denoising task in Section 4.2 and show that its $\gamma$-loss can be expressed as a sum of an $\ell_1$ reconstruction loss and a $\gamma$-power regularizer.

**Problem setup** We first describe the problem setting for the denoising task.. This setting is analogous to the denoising criterion in Im et al. (2016), which established an explicit loss formulation for Gaussian VAEs under denoising corruption

Let $x \sim p_{\text{data}}$ denote data samples, and let $x' \sim r(x'|x)$ denote corrupted samples, where $r(x'|x)$ is the corruption distribution (e.g. $x' = x + \epsilon$ with $\epsilon \sim p_{\text{noise}}$, $x \perp \epsilon$). The corrupted input $x'$ is fed into the VAE, consisting of a conditional encoder $q_\phi(z|x')$ and decoder $p_\theta(x|z)$, yielding a reconstruction output $\hat{x}$.

Our goal is to recover $x$ from $x'$, thereby achieving a denoising effect. To formalize this setting, we impose a Markovian assumption under which the following conditional independencies hold, as depicted in Figure 5:

$$x \perp z \mid x', \quad x' \perp \hat{x} \mid z.$$

Analogous to (7), we consider two joint manifolds over $(x, z)$, where the effect of the corrupted input $x'$ is absorbed into $\tilde{q}_\phi(z \mid x)$:

$$\mathcal{M}_{\text{model}} = \{p_\theta(x, z) = p_\theta(x|z)\, p(z) : \theta \in \Theta\},$$

$$\mathcal{M}_{\text{data}} = \{q_\phi(x, z) = p_{\text{data}}(x)\, \tilde{q}_\phi(z|x) : \phi \in \Phi\}, \quad \tilde{q}_\phi(z|x) = \int q_\phi(z|x')r(x'|x)dx'.$$

We then formulate the joint minimization problem for the $\gamma$-loss, with the goal of deriving its explicit form:

$$\underset{\phi \in \Phi, \theta \in \Theta}{\arg\min}\, D_\gamma(q_\phi(x, z) \,\|\, p_\theta(x, z)).$$

**Derivation of the $\gamma$-loss for the denoising task** First, the $\gamma$-power cross entropy is computed as follows:

$$
\begin{aligned}
\mathcal{C}_\gamma(q_\phi(x, z), p_\theta(x, z)) &= -\int p_{\text{data}}(x)\tilde{q}_\phi(z|x) \frac{p_\theta(x, z)^\gamma}{\|p_\theta(x, z)\|_{1+\gamma}^\gamma} dz\, dx \\
&= -\int p_{\text{data}}(x) \left(\int q_\phi(z|x')r(x'|x)dx'\right) \frac{p_\theta(x, z)^\gamma}{\|p_\theta(x, z)\|_{1+\gamma}^\gamma} dz\, dx \\
&= -\int\int p_{\text{data}}(x)q_\phi(z|x') \frac{p_\theta(x, z)^\gamma}{\|p_\theta(x, z)\|_{1+\gamma}^\gamma} r(x'|x)dz\, dx'\, dx \\
&= -\mathbb{E}_{x \sim p_{\text{data}}}\mathbb{E}_{x' \sim r(x'|x), z \sim q_\phi(z|x')} \left[\frac{p_\theta(x, z)^\gamma}{\|p_\theta(x, z)\|_{1+\gamma}^\gamma}\right].
\end{aligned}
$$

For the $\gamma$-power entropy, define $\sigma_\phi(x)$ by $\mathcal{H}_\gamma(\tilde{q}_\phi(\cdot|x)) = -\sigma_\phi(x)^{-\frac{\gamma}{1+\gamma}}$. Applying Proposition B.1, we obtain

$$\mathcal{H}_\gamma(q_\phi(x,z)) = -\left(\int p_{\text{data}}(x)^{1+\gamma}\left(\int \tilde{q}_\phi(z|x)^{1+\gamma}\,dz\right)dx\right)^{\frac{1}{1+\gamma}}$$

$$= -\int p_{\text{data}}(x)\,\sigma_\phi(x)^{-\frac{\gamma}{1+\gamma}}\,dx + \gamma\mathcal{H}(p_{\text{data}}) + O(\gamma^2).$$

Substituting $\mathcal{H}_\gamma(\tilde{q}_\phi(\cdot|x))$ into the approximation yields

$$\mathcal{H}_\gamma(q_\phi(x,z)) = \int p_{\text{data}}(x)\,\mathcal{H}_\gamma(\tilde{q}_\phi(\cdot|x))\,dx + \gamma\int p_{\text{data}}(x)\log p_{\text{data}}(x)\,dx + O(\gamma^2)$$

$$= \mathbb{E}_{x\sim p_{\text{data}}}\left[\mathcal{H}_\gamma(\tilde{q}_\phi(\cdot|x))\right] + \gamma\mathcal{H}(p_{\text{data}}) + O(\gamma^2).$$

Since $p_\theta(x,z)$ is a joint symPareto distribution and $\tilde{q}_\phi(z\mid x)$ remains in the conditional symPareto family after averaging over the corruption process, the resulting divergence has the similar form as in (7):

$$\mathcal{L}_\gamma(\theta,\phi) = \mathbb{E}_{x\sim p_{\text{data}}}\mathbb{E}_{x'\sim r(\cdot|x)}\left[\frac{1}{\sigma}\mathbb{E}_{z\sim q_{\phi,\nu}(\cdot|x')}\|x-\mu_\theta(z)\|_1 + 2\alpha D_\gamma(q_{\phi,0}\parallel p_{\text{alt}}) + 2\alpha\beta\|\mu_\phi(x)\|_1\right],$$

The only modification compared to the standard $\gamma$-loss (12) is the additional expectation over the corrupted input $x'\sim r(\cdot|x)$.

## D  DISCUSSION

**Computation cost for training**  In contrast to models that involve numerical integration or iterative estimation of tail parameters, ParetoVAE leverages the closed-form expression of the $\gamma$-loss and the simple Laplace–Gamma mixture reparametrization of the symPareto distribution. This construction avoids computational bottlenecks and ensures stable training dynamics. Although the reparameterization involves one additional sampling step compared to light-tailed VAEs (e.g., Gaussian VAE or Laplace VAE), the runtime of ParetoVAE remains virtually identical to these models.

**Tuning hyperparameter $\nu$**  Recent VAE models based on Student's $t$ distributions (Kim et al., 2024; Bouayed et al., 2025) highlight the challenge of tuning the hyperparameter $\nu$. ParetoVAE encounters a similar issue. Previous works have shown empirically that selecting $\nu$ from a suitable range (e.g., $\nu \in [2.5, 20]$ in (Bouayed et al., 2025)) is sufficient for stable performance in high-dimensional settings, although this criterion lacks theoretical guarantees. Abiri & Ohlsson (2020) proposes learning $\nu$ in the latent space, but learning $\nu$ may require additional constraints or gradient estimators; in our preliminary attempts with additional structures to learn $\nu$, we did not observe consistent improvements.

To illustrate how performance depends on $\nu$, we vary the tail parameter $\nu$ over a wide range and report the resulting PSNR values in Table 5.

To keep this appendix concise, we report PSNR only; SSIM and accuracy exhibit the same qualitative trend. We also note that $t^3$VAE requires $\nu > 2$, whereas ParetoVAE only requires $\nu > 1$.

Table 5: PSNR sensitivity to the tail parameter $\nu$.

| Model | $\nu = 1.1$ | $\nu = 2.1$ | $\nu = 3.1$ | $\nu = 5.1$ | $\nu = 10.1$ |
|---|---|---|---|---|---|
| ParetoVAE | 24.268 | 23.424 | 22.670 | 17.590 | 19.443 |
| $t^3$VAE | – | 17.812 | 17.878 | 19.403 | 19.044 |

| Model | $\nu = 15.1$ | $\nu = 20.1$ | $\nu = 25.1$ | $\nu = 50.1$ | $\nu = 100.1$ |
|---|---|---|---|---|---|
| ParetoVAE | 17.590 | 12.064 | 11.991 | 12.041 | 12.047 |
| $t^3$VAE | 19.294 | 20.652 | 18.931 | 17.329 | 16.263 |

In our setup, ParetoVAE attains its best performance at moderate $\nu$ within a relatively narrow range compared to Student's $t$-based VAEs. A plausible explanation is that symPareto already induces sufficiently heavy tails for relatively small $\nu$, so the robustness benefit saturates over a shorter interval. Moreover, the $\ell_1$-based structure improves stability against sparse large deviations, further reducing the need to search over a wide range of $\nu$. Nevertheless, there is still no theoretical foundation for optimal $\nu$ selection, and we regard this as an important open problem in the study of heavy-tailed generative models.

**Training strategies for regularizer** Balancing the individual terms of a combined loss function (reconstruction + regularizer) is a key challenge in training VAE-style models, as exemplified by methods such as (Higgins et al., 2017) and (Fu et al., 2019). Similarly, it is important for ParetoVAE to achieve an appropriate balance between the reconstruction and regularization terms. To this end, we explore various scheduling methods and empirically found that adjusting the regularization weight with a logistic or linear function is more effective than using cyclic or cosine annealing. This effect is particularly evident when combining loss terms with disparate properties, such as an $\ell_1$-based reconstruction term with an $\ell_2^2$-based regularization term, or vice versa.

