# OpenReview forum: "Pareto Variational Autoencoder"
_ICLR.cc/2026/Conference — ICLR 2026 Poster_

### Official Review · Reviewer_iLe7 · 2025-10-30

**Soundness:** 3
**Presentation:** 3
**Contribution:** 3
**Rating:** 6
**Confidence:** 3

**Summary:**

This paper proposes a novel VAE based on the information geometry of $\gamma$-divergence together with a multivariate symmetric Pareto (symPareto) distribution. Since the symPareto distribution is heavy-tailed and promotes sparsity, it improves both the performance and robustness of VAEs. A variety of experiments validate the proposed method.

**Strengths:**

Introducing complex distributions into VAEs is difficult because one must compute the KL divergence between the encoder and the prior and also sample from the encoder via the reparameterization trick. This paper incorporates a complex distribution like symPareto into a VAE in a theoretically sound and tractable way. The empirical results are also strong.

**Weaknesses:**

Please see the Questions section.

**Questions:**

- In the proposed VAE, symPareto is used for the encoder and the prior, and either a t-distribution (L2) or symPareto (L1) is used for the decoder. Which component is particularly effective? For example, if the goal is to induce sparsity in the latent variables, using only a symPareto prior may be sufficient. If the KL can be computed, it even seems acceptable to keep the encoder Gaussian. On the other hand, when observations are noisy or sparsity is required, using a t-distribution or symPareto for the decoder appears effective. For the image denoising experiment, for instance, it seems a t-decoder should handle the task; yet the proposed method achieves higher performance than t-based VAEs on all datasets. Where does this performance gain come from?
- Using toy data, can you visualize how the latent variables behave under a symPareto prior? For example, as in Figure 6 of [1], train on a four-dimensional one-hot vector dataset and visualize the two-dimensional latent variables. I believe this would make the effect of the symPareto prior easier to understand.
- VAEs often suffer performance degradation due to over-regularization by the prior. Does employing a symPareto prior resolve this issue? Moreover, it is known that the aggregated posterior (a infinite mixture distribution of the encoder) can address this problem [2]. Could your method be combined with such approaches?

[1] Mescheder, Lars, Sebastian Nowozin, and Andreas Geiger. "Adversarial variational bayes: Unifying variational autoencoders and generative adversarial networks." International conference on machine learning. PMLR, 2017.

[2] Tomczak, Jakub, and Max Welling. "VAE with a VampPrior." International conference on artificial intelligence and statistics. PMLR, 2018.

---

> ### Author Response · Authors · 2025-11-20
> **Response to Reviewer iLe7 [1/2]**
>
> Thank you for the thoughtful and technically insightful questions. Your questions helped us clarify the theoretical motivations and empirical behavior of ParetoVAE. Below, we provide detailed responses to each point.
>
> ### [Q1] In the proposed VAE, symPareto is used for the encoder and the prior, and either a t-distribution (L2) or symPareto (L1) is used for the decoder. Which component is particularly effective? Where does this performance gain come from?
>
> **Response:**
>
> ### **1. Encoder / Prior selection**
>
> First, KL divergence between Gaussian encoder and Pareto prior does not admit a closed form due to the second term.
>
> $$
> D_{KL}(N_n(z|\mu_\phi(x), \sigma^2(x)I_n) || P_n(z|0,1,\nu)) = -H(N(\mu_\phi(x), \sigma_\phi(x)^2I)) - (\nu+n)E_{Z \sim N_n(\mu_\phi(x), \sigma_\phi(x)^2I_n)}[\log(1+ \frac{||Z||_1}{\nu})] + const
> $$
>
> The same issue arises for the $\gamma$-power loss: especially when computing $\gamma$-power cross-entropy:
>
> $$
> \int p(x) q(x)^rdx \propto \int \exp\left(-\frac{||x-\mu||_2^2}{2\sigma^2}\right) \left(1 +\frac{||x||_1}{\nu}\right)^{-(\nu+n)} dx
> $$
>
> Although one may approximate these terms via Monte Carlo, doing so incurs high variance, especially under heavy-tailed densities. Hence it becomes computationally expensive as many samples are needed for stability.
>
> For these reasons, our framework uses matched heavy-tailed families (symPareto–symPareto), which allows closed-form expressions under the $\gamma$-power divergence.
>
> ### **2. Decoder selection**
>
> As for the decoder, its choice determines the reconstruction loss: symPareto induces an L1-type loss, whereas the t-decoder yields an L2-type loss. This decoder selection can be interpreted through the lens of classical regression, where L1 and L2 losses correspond to median and mean regression respectively. From the perspective of robust M-estimation, it is well established that L1-type estimators offer improved robustness to heavy-tailed noise and outliers compared to their L2-based counterparts [1].
>
> This insight carries directly into our generative setting and explains ParetoVAE’s improved denoising performance: the symPareto decoder suppresses the influence of corrupted pixels more effectively than the t-decoder, yielding higher robustness even when both models incorporate heavy-tailed components.
>
> [1] P. J. Huber. Robust Statistics. Wiley Series in Probability and Statistics, 1981.
>
> ### [Q2] Using toy data, can you visualize how the latent variables behave under a symPareto prior?
>
> **Response:** We first note that the OpenReview system does not allow uploading additional figures within the rebuttal interface. Since we plan to submit a full revised manuscript at the end of the rebuttal period, we cannot attach the latent plots immediately. Instead, we summarize the experimental setup and the qualitative behavior we observed.
>
> Following your suggestion, we trained VAEs on a simple 4-dimensional heavy-tailed distribution, $t_4(0,I_4,\nu)$, using the same priors as in our previous experiments: Gaussian, Laplace, Student’s $t$, and symPareto (with fixed $\nu=10$ for both heavy-tailed models). We then separated normal and outlier samples using a norm threshold. For visualization, we set the latent dimension to 2.
>
> The Gaussian and Laplace priors produce latent clusters that remain concentrated near the center, even for outlier samples. In contrast, the Student’s $t$ and symPareto priors yield much clearer separation than their light-tailed counterparts. With the $t$ prior, the latent space becomes more elliptical, causing outliers to spread farther from the center. The symPareto prior, on the other hand, allocates a broader latent region, and its $\ell_{1}$-type geometry pulls normal samples tightly toward their class centers while pushing outliers toward cross-shaped extremal directions. As a result, both heavy-tailed priors provide substantially more interpretable latent organization, with symPareto offering the most distinct boundary structure among the four.

---

> > ### Author Response · Authors · 2025-11-20
> > **Response to Reviewer iLe7 [2/2]**
> >
> > ### [Q3] VAEs often suffer performance degradation due to over-regularization by the prior. Does employing a symPareto prior resolve this issue? Moreover, it is known that the aggregated posterior (a infinite mixture distribution of the encoder) can address this problem [2]. Could your method be combined with such approaches?
> >
> > **Response:** From a Bayesian perspective, the symPareto prior admits the following scale–mixture representation:
> >
> > $P\_n(x|0, 1_n, \nu) = \int L\_n(x|0, \nu \lambda^{-1} I\_n) Gamma(\lambda | \nu, 1) d\lambda,$
> >
> > where $\mathcal{L}_n(x|0,\nu \lambda^{-1}I_n)$ denotes the product Laplace densities, obtained as the product of $n$ independent univariate Laplace distributions with scale parameter $\nu \lambda^{-1}$, and  $Gamma(\lambda | \nu, 1)$ denotes the Gamma density with shape parameter $\nu$ and rate parameter $1$. This form is analogous to the classical Gaussian–chi-square mixture representation of the Student’s $t$ distribution. This shows that the symPareto prior can be interpreted as an “infinite” scale–mixture of Laplace distributions.
> >
> > Although this differs structurally from the Gaussian mixture form of the aggregated posterior in [2], both mechanisms share a key property: they introduce mixture components and consequently relax the rigidity imposed by a single light-tailed prior. In this sense, the symPareto prior provides a principled heavy-tailed alternative capable of mitigating over-regularization, which is consistent with the improvements observed in $t^3$VAE as well as in our own experiments.
> >
> > Furthermore, regarding your suggestion on combining ParetoVAE with aggregated-posterior approaches, we have explored the construction of an “aggregated symPareto posterior.” However, incorporating the $\gamma$-power divergence into such mixtures leads to analytically intractable expressions, and our initial attempts did not result in a useful formulation. We consider this a promising direction for future work and plan to investigate tractable approximations or relaxations.
> >
> > [2] Tomczak, Jakub, and Max Welling. "VAE with a VampPrior." International conference on artificial intelligence and statistics. PMLR, 2018.

---

> > > ### Comment · Reviewer_iLe7 · 2025-11-27
> > >
> > > Thank you for your response. I will keep the score at 6, but I will increase my confidence.

---

### Official Review · Reviewer_gNb7 · 2025-10-30

**Soundness:** 3
**Presentation:** 3
**Contribution:** 3
**Rating:** 6
**Confidence:** 3

**Summary:**

The manuscript discusses changing enforcing the Gaussianity of the distribution in the latent space of the variational autoencoders (VAE).
The manuscript proposes  ParetoVAE, a probabilistic autoencoder that minimizes the γ-power, instead of the Gaussian assumption as in the classical VAE, divergence between two statistical manifolds. Also Student's t and symPareto distributions are considered. These types of VAEs show good results in sparse, heavy-tailed data reconstruction and word frequency analysis.
The symPareto decoder enables robust high-dimensional denoising.

**Strengths:**

- The manuscript aims to develop a VAE that would be more appropriate for certain types of data.
- The corresponding loss function for the new VAE is derived.
- An application seems to be in denoising, when images are corrupted by Salt and Pepper noise

**Weaknesses:**

- Lack of theoretical development to justify why the new distributions are necessary in certain data representations. While the manuscript provides statistical distributions known in statistical analysis, it does not connect these with data reconstruction.
- Experimental results are rather limited, especially in real data.

**Questions:**

What is the performance of the ParetoVAE in the case when representing images corrupted by Gaussian noise?

More references should have been provided for the statistical derivations from the Appendices, in the supplementary material.

---

> ### Author Response · Authors · 2025-11-20
> **Response to Reviewer gNb7**
>
> Thank you for the thoughtful and constructive feedback on both the theoretical development and the empirical analysis. Rather than submitting a partial revision during the rebuttal period, we use this response to outline the updates we plan to incorporate. These revisions will be further refined and integrated into the final manuscript after internal discussion among the authors.
>
> ### [W1] Lack of theoretical development to justify why the new distributions are necessary in certain data representations. While the manuscript provides statistical distributions known in statistical analysis, it does not connect these with data reconstruction.
>
> **Response:** In the classical extreme value theory, the Pickands–Balkema–De Haan theorem [1,2] characterizes the generalized Pareto distribution as the canonical limit for threshold exceedances. The marginal of the proposed symPareto distribution reduces to a generalized Pareto distribution. We believe this connection provide a theoretical ground for employing Pareto-type components when modeling data that exhibit heavy-tail or extreme value behaviors. We will incorporate this point in the revision.
>
> [1] J. Pickands. Statistical Inference Using Extreme Order Statistics. The Annals of Statistics, 1975.
>
> [2] A. Balkema and L. Haan. Residual life time at great age. Annals of Probability, 1974.
>
> ### [W2] Experimental results are rather limited, especially in real data.
>
> **Response:** Due to limited computational resources, we were unable to conduct experiments on datasets as large as ImageNet. Nevertheless, the present results indicate that ParetoVAE handles realistic image resolutions and complexity effectively, suggesting that its strengths would generalize to even larger datasets. While such large-scale experiments could not be completed in time, we will make further attempts during the remainder of the rebuttal period to include additional results if feasible.
>
> ### [Q1] What is the performance of the ParetoVAE in the case when representing images corrupted by Gaussian noise?
>
> **Response:** We conducted additional experiments on images corrupted by Gaussian noise (clipped to the normalized pixel range [0,1], varying the noise level with standard deviations ($\sigma =0.01$ and $\sigma = 0.5$). For brevity, we report PSNR scores, though the other metrics (SSIM, accuracy) show a similar trend. The results show that ParetoVAE consistently outperforms the baselines even under Gaussian noise. This behavior is reasonable: Gaussian noise on $[0,1]$ behaves as a continuous, smoothed counterpart of salt-and-pepper noise, and the robustness induced by the ParetoVAE carries over to this setting as well.
>
> | Model | MNIST (0.01 / 0.5) | CelebA (0.01 / 0.5) | SVHN (0.01 / 0.5) | CIFAR-10 (0.01 / 0.5) | Omniglot (0.01 / 0.5) |
> | --- | --- | --- | --- | --- | --- |
> | ParetoVAE | 24.631±0.186 / 22.224±0.205 | 24.369±0.069 / 23.983±0.183 | 24.343±0.070 / 26.647±0.825 | 20.386±0.776 / 20.535±0.222 | 20.831±0.063 / 19.657±0.115 |
> | LVAE | 17.853±0.028 / 17.480±0.277 | 18.930±0.106 / 18.805±0.070 | 21.055±0.105 / 20.901±0.396 | 15.833±0.214 / 15.816±0.128 | 11.962±0.061 / 11.963±0.062 |
> | t3VAE | 19.962±0.174 / 18.245±0.172 | 18.797±0.019 / 18.778±0.219 | 22.961±0.258 / 21.871±0.603 | 16.244±0.203 / 16.002±0.058 | 11.960±0.059 / 11.960±0.059 |
> | VAE | 12.492±0.263 / 12.367±0.236 | 14.245±0.056 / 14.207±0.145 | 20.019±0.125 / 20.033±0.057 | 15.918±0.218 / 15.811±0.348 | 11.960±0.064 / 11.958±0.062 |
>
>
> ### [Q2] More references should have been provided for the statistical derivations from the Appendices, in the supplementary material.
>
> **Response:** Thank you for the helpful suggestion. Following your advice, we have carefully reviewed all derivations in the Appendices and plan to add appropriate references to support the statistical results in the revision. In particular, we will cite Scheffé’s theorem [1] for convergence arguments and standard results on spherical symmetry showing that covariances between distinct components vanish [2]. Additional references will also be included where relevant throughout the supplementary derivations.
>
> [1]  H. Scheffe. A Useful Convergence Theorem for Probability Distributions. The Annals of Mathematical Statistics, 1947.
>
> [2] K. Fang, S. Kotz, and K. Ng. Symmetric Multivariate and Related Distributions. Classics in Applied Mathematics, 2017.

---

### Official Review · Reviewer_dmgt · 2025-10-31

**Soundness:** 4
**Presentation:** 3
**Contribution:** 4
**Rating:** 8
**Confidence:** 4

**Summary:**

The paper introduces the Pareto variational auto encoder. The model addresses the problem of modeling heavy tailed data within the VAE framework, which most typically use exponential family distributions. The paper offers a detailed and clear mathematical exposition for the multivariate Pareto distribution, uses information geometric tools to construct gamma flat manifold to analyze gamma power divergence, and presents detailed simulation results comparing to relevant baseline models on data with heavy tailed structure including graphs and language.

**Strengths:**

Strengths of this paper include:
- The mathematical exposition was very nice.
- The core motivation of modeling heavy tailed data was clearly borne out in experiments
- The proposed model raised interesting challenges that were well resolved mathematically.
- The approach was compared with a set of baselines that gave a clear sense of why the Pareto model was succeeding

**Weaknesses:**

Weaknesses include:
- The introduction could have been more concrete sooner. Many of the arguments in the introduction would have been stronger if grounded in specific examples of data that are heavy tailed, specifically which models are under consideration, etc. The remainder of the paper answered these questions well.


Detailed comments:
- "Incorporating robustness in generative modeling has enticed many researchers
of the field." This sentence is not necessary.
- "To this end, ..." To the end of enticing researchers into the field?
- "the γ-power divergence that naturally populates power-law families" what does populate mean here?
- "Conventional VAEs typically employ exponential-family distributions, most notably the Gaussian, for their probabilistic model due to their mathematical tractability." This is typical but certainly not universally true. Therefore the next sentence "To address these limitations," doesn't follow. It is necessary to list the exceptions to the convention and argue why they don't address the problem you want to solve.
- "exhibit heavy tails and extreme events" What are the events you have in mind here? Is there evidence that standard VAEs haven't succeeded in modeling these events? Have modelers just not tried to use VAEs?
- What do you mean precisely for a numerical integration to be intractable? (Line 044)
- Table 1 is not particularly informative. Is there something important that I am missing there?
- typo: exponentiall
- typo: includde
- Two papers that are not cited but may be of interest are the InfoVAE and Coupled VAE (cites below) which connect to entropic OT (itself connected to Information Geometry).

Zhao, S., Song, J., & Ermon, S. (2017). Infovae: Information maximizing variational autoencoders. arXiv preprint arXiv:1706.02262.
Hao, X., & Shafto, P. (2023). Coupled variational autoencoder. arXiv preprint arXiv:2306.02565.

**Questions:**

I don't have major questions. I enjoyed the paper, thanks!

---

> ### Author Response · Authors · 2025-11-20
> **Response to Reviewer dmgt [1/2]**
>
> Thank you for the thorough and constructive feedback, especially for your close reading of intro sections and your detailed suggestions on how to improve the clarity of the abstract and introduction. Rather than immediately uploading a full revised manuscript during the rebuttal period, we use this response to outline the concrete revision plan. For the detailed points below, we have already drafted specific textual changes, and we will further refine them in discussion among the authors before submitting the final version.
>
> ### [Abstract]
>
> [A1] "Incorporating robustness in generative modeling has enticed many researchers of the field." This sentence is not necessary.
>
> [A2] "To this end, ..." To the end of enticing researchers into the field?
>
> [A3] "the γ-power divergence that naturally **populates** power-law families" what does populate mean here?
>
> **Response:** Following your suggestion, we have revised the abstract as follows:
>
> This paper introduces a new class of multivariate power-law distributions---the symmetric Pareto (symPareto) distribution---which can be viewed as an $\ell_1$-norm-based counterpart of the multivariate $t$ distribution, with the motivation of capturing the heavy tail of the target distribution in generative modeling and bringing robustness to noise in downstream tasks such as image denoising. The symPareto distribution possesses many attractive information-geometric properties with respect to the $\gamma$-power divergence that is a natural alternative to the Kullback-Leibler divergence, the core of the conventional variational autoencoder (VAE) models, for power families. Leveraging on the joint minimization view of variational inference, this paper proposes the ParetoVAE, a probabilistic autoencoder that minimizes the $\gamma$-power divergence between two statistical manifolds. ParetoVAE employs the symPareto distribution for both prior and encoder, with flexible decoder options including multivariate $t$ and symPareto distributions. Empirical evidences demonstrate the effectiveness of ParetoVAE across multiple domains through varying the types of the decoder.
> The $t$ decoder achieves superior performance in sparse, heavy-tailed data reconstruction and word frequency analysis; the symPareto decoder enables robust high-dimensional denoising.

---

> > ### Author Response · Authors · 2025-11-20
> > **Response to Reviewer dmgt [2/2]**
> >
> > ## [Introduction]
> >
> > ### [I1] and [I2]
> > [I1] "Conventional VAEs typically employ exponential-family distributions, most notably the Gaussian, for their probabilistic model due to their mathematical tractability." This is typical but certainly not universally true. Therefore the next sentence "To address these limitations," doesn't follow. It is necessary to list the exceptions to the convention and argue why they don't address the problem you want to solve.
> >
> > [I2] What heavy-tailed/extreme events do you have in mind? Is there evidence that standard VAEs fail to model them? Or have people simply not tried VAEs?
> >
> > **Response:** In our revision, we plan to clarify both (i) the types of heavy-tailed or extreme behaviors we target, and (ii) why standard Gaussian-based VAEs are structurally insufficient for these regimes.
> >
> > First, many real-world datasets exhibit pronounced heavy-tailed or power-law behavior—for example, scale-free network degree distributions [1] and long-tailed class-frequency distributions in large-scale classification tasks [2]. Such patterns substantially violate Gaussian assumptions. Prior work further reports that Gaussian-based VAEs under-estimate tail probabilities, over-regularize latent codes, and fail to capture rare but informative events in these settings [3,4]. We will highlight these documented limitations to clarify that the issue stems from an inherent mismatch between light-tailed VAE components and heavy-tailed data.
> >
> > In addition to clarifying these limitations in the revision, we will briefly position our model in relation to prior work. Existing Student’s t-based VAEs [4–6] represent previous attempts to address such heavy-tailed or extreme behaviors. Our approach follows the same motivation but explores a different, theoretically grounded heavy-tailed family: classical extreme-value theory highlights Pareto-type distributions as canonical for tail and exceedance modeling [7,8]. This motivates our use of the symPareto distribution as a complementary alternative to t-based VAEs.
> >
> > [1] A.L. Barabási and R. Albert. “Emergence of scaling in random networks”. Science, 1999.
> >
> > [2] Y.Zhang, B. Kang, B., S. Yan, and J. Feng., Deep long-tailed learning: A survey. IEEE TPAMI, 2023.
> >
> > [3] N. Lafon, P. Naveau, and R. Fablet. A VAE Approach to Sample Multivariate Extremes, Arxiv, 2023
> >
> > [4] J. Kim, J, Kwon, M. Cho, H. Lee, and J. Won. $t^3$-variational autoencoder: Learning heavy-tailed data with student’s t and power divergence. ICLR, 2024.
> >
> > [5] H. Takahashi, T. Iwata, Y. Yamanaka, M. Yamada, and S. Yagi. Student-t variational autoencoder for robust density estimation. IJCAI, 2018
> >
> > [6] N. Abiri and M. Ohlsson. Variational auto-encoders with student’s t-prior, Arxiv, 2020
> >
> > [7] J. Pickands. Statistical Inference Using Extreme Order Statistics. The Annals of Statistics, 1975.
> >
> > [8] A. Balkema and L. Haan. Residual life time at great age. Annals of Probability, 1974.
> >
> > ### [I3] What do you mean precisely for a numerical integration to be intractable?
> >
> > **Response:** Our intended meaning is that the KL divergence between two symPareto distributions has no closed-form expression. Consequently, evaluating this KL term requires computing high-dimensional integrals involving the symPareto density, for which no analytical simplification exists. Such integrals become computationally prohibitive as the dimension grows, making direct numerical integration effectively infeasible and requiring additional approximation methods such as Monte Carlo, at the cost of extra variability (see our response to W2 of Reviewer gUPd). We will clarify this point explicitly in the revised manuscript.
> >
> > ### [I4] Table 1 is not particularly informative. Is there something important that I am missing there?
> >
> > **Response:** Table 1 is intended to provide a first-glance summary of heavy-tailed VAE variants, highlighting how introducing the symPareto family expands the range of latent/decoder distribution combinations beyond those available in $t^3$VAE. To clarify its purpose, we will add a brief explanatory sentence in the caption and main text.
> >
> > ### [I5] typo: exponentiall (Line 154), includde (Line 351)
> > **Response:** We will correct these typos. Thank you for pointing them out.
> >
> > ### [I6] Two papers that are not cited but may be of interest are the InfoVAE and Coupled VAE (cites below) which connect to entropic OT (itself connected to Information Geometry).
> >
> > **Response:** Thank you for the helpful pointers. We reviewed both InfoVAE and Coupled VAE and found that they share conceptual similarities with our joint minimization view of the VAE objective. Given this connection, we will add these works to the related-work section to better situate our contribution within the broader landscape of alternative objectives to the ELBO.

---

> > > ### Comment · Reviewer_dmgt · 2025-11-21
> > >
> > > Thanks for the detailed response!

---

### Official Review · Reviewer_gUPd · 2025-10-31

**Soundness:** 3
**Presentation:** 3
**Contribution:** 3
**Rating:** 6
**Confidence:** 3

**Summary:**

This paper introduces the symmetric multivariate Pareto distribution (symPareto) and leverages it to build ParetoVAE, a new class of heavy-tailed variational autoencoders. The authors motivate the need for power-law latent distributions when modeling real-world data with extreme values or rare events, addressing well-known limitations of Gaussian-based VAEs. The work is grounded in a solid information-geometric perspective: the γ-power divergence is employed to derive a tractable VAE objective, circumventing difficulties associated with ELBO estimation for power-law distributions. Empirical studies across multiple domains (including noisy image reconstruction and denoising tasks) demonstrate improved robustness to outliers, better sparse-data handling, and enhanced recovery of fine-grained features relative to baseline VAEs and heavy-tailed t-VAEs. Overall, this paper makes a meaningful and well-motivated contribution to the study of robust generative modeling.

**Strengths:**

1. Strong conceptual motivation for replacing exponential-family assumptions with heavy-tailed distributions in real-world generative modeling.

2. Novel modeling component: introduction of the symPareto distribution as a multivariate power-law prior/encoder, extending beyond the Student’s-t VAE line.

3. Theoretically grounded via γ-power divergence and joint minimization formulation, providing a tractable objective.

4. Empirical benefits validated across several benchmarks, especially in high-noise and heavy-tailed settings. and Practical insights into when symPareto vs Student’s-t decoders are beneficial.

5. Clear writing and strong experimental motivation, making the work accessible despite its mathematical basis.

**Weaknesses:**

1. Computational trade-offs not fully quantified: while the γ-loss improves tractability, a more detailed cost vs. ELBO comparison would help practitioners.

2. Ablations on γ-power choices and sensitivity analysis would improve clarity on hyperparameter robustness.

**Questions:**

Please see above weakness for questions to answer.

---

> ### Author Response · Authors · 2025-11-20
> **Response to Reviewer gUPd**
>
> Thank you for your insightful comments on the practical aspects of our $\gamma$-loss framework, including computational trade-offs, the need for ablations on the $\gamma$-power divergence, and hyperparameter sensitivity. Below, we provide our detailed responses to each point.
>
> ### [W1] Computational trade-offs not fully quantified: while the $\gamma$-loss improves tractability, a more detailed cost vs. ELBO comparison would help practitioners.
>
> **Response:**  To clarify this point, let us compare the cost of ELBO-based training with that of our proposed $\gamma$-loss under symPareto prior/encoder with a $t$-decoder (L2). For ParetoVAE, the ELBO decomposes into three parts:
>
> $\mathrm{ELBO}(x) = \mathbb{E}\_{q\_\phi(z|x)}\[\log p\_\theta(x|z)\] - \mathbb{E}\_{q\_\phi(z|x)}\[\log q\_\phi(z|x)\] + \mathbb{E}\_{q\_\phi(z|x)}[\log p(z)].$
>
> The first term yields a logarithmic reconstruction loss of the form:
>
> $\mathbb{E}\_{q\_\phi(z|x)}\[\log p\_\theta(x|z)\] \propto \mathbb{E}\_{z \sim q_\phi(\cdot|x)}\left[\log \left(1+\frac{\lVert x-\mu\_\phi(z)\rVert\_2^2}{(\nu+m)\sigma^{2}}\right)\right].$
>
> The second term $\mathbb{E}\_{q\_\phi(z|x)}\[\log q\_\phi(z|x)\] $ admits a closed form.
>
> $\mathbb{E}\_{q\_\phi(z|x)}\[\log q\_\phi(z|x)\] = -\frac{2(\nu+n+m)}{(\nu+n-1)^2}C\_{n,\nu+n,\nu+n} - \sum_{i=1}^m \log \sigma\_{\phi,i}(x) + const.$
>
> The third term has no closed-form expression and therefore must be approximated using Monte Carlo sampling, which is a source of extra variability. The variance of the Monte Carlo estimate is large since the sampling distribution is heavy-tailed.
>
> $\mathbb{E}\_{z \sim q\_\phi(\cdot|x)}[\log p(z)] \approx \frac{1}{K}\sum_{j=1}^K \log p(z^{(j)}), \quad z^{(j)} \sim q\_\phi(\cdot|x)$
>
> To quantify the computational impact of this term, we measured the wall-clock time required for loss evaluation on MNIST (batch size 128), normalizing the $\gamma$-loss runtime to 1.0. The results for latent dimensions 32 and 64 are summarized below.
>
> | Model (latent dim = 32) | Relative Runtime | Model (latent dim = 64) | Relative Runtime |
> | --- | --- | --- | --- |
> | ParetoVAE (γ-loss) | 1.00 | ParetoVAE (γ-loss) | 1.00 |
> | ParetoVAE (ELBO, K=1) | 1.02 | ParetoVAE (ELBO, K=1) | 1.04 |
> | ParetoVAE (ELBO, K=4) | 1.06 | ParetoVAE (ELBO, K=4) | 1.33 |
> | ParetoVAE (ELBO, K=16) | 1.24 | ParetoVAE (ELBO, K=16) | 1.40 |
>
> While there is almost no difference at $K=1$, the gap grows steadily as either $K$ or the latent dimension increases. This observation further accentuates the computational advantage of the $\gamma$-loss formulation.
>
> ### [W2] Ablations on $\gamma$-power choices and sensitivity analysis would improve clarity on hyperparameter robustness.
>
> **Response:** Regarding the $\gamma$-loss, we note that its three components: (reconstruction error, $\gamma$-power divergence, and the $\ell\_1$ term on $\mu\_\phi$), arise directly from the theoretical joint-minimization formulation and theorem 2.1.
>
> These terms are therefore structurally necessary rather than heuristic design choices. In practice, ablating any one of the three components results in training failure (e.g., collapsed latent representations or divergence during optimization). Since these outcomes are trivial and uninformative, we omit the ablation results.
>
> For the tail parameter $\nu$, we conducted a sensitivity analysis by varying $\nu$ over a wide range and found that ParetoVAE remains stable across a broad interval. To illustrate this behavior concisely, we report PSNR scores only and omit the other metrics (SSIM and accuracy), as they exhibit the same trend.
>
> | Model | $\nu = 1.1$ | $\nu = 2.1$ | $\nu = 3.1$ | $\nu = 5.1$ | $\nu = 10.1$ |
> | --- | --- | --- | --- | --- | --- |
> | ParetoVAE | 24.26822 | 23.42383 | 22.66968 | 17.5903 | 19.44277 |
> | t3VAE | - | 17.81159 | 17.87794 | 19.40275 | 19.04375 |
> | Model | $\nu = 15.1$ | $\nu = 20.1$ | $\nu = 25.1$ | $\nu = 50.1$ | $\nu = 100.1$ |
> | ParetoVAE | 17.5903 | 12.06405 | 11.99119 | 12.04142 | 12.04738 |
> | t3VAE | 19.29435 | 20.65228 | 18.93076 | 17.32915 | 16.26345 |
>
> (Note that $t^3$VAE requires $\nu > 2$, whereas ParetoVAE only requires $\nu > 1$)
>
> As discussed in Appendix D, there is currently no universally accepted rule for selecting $\nu$. Nevertheless our experiments suggest a wide operating range in which ParetoVAE performs well. Developing a more principled tuning strategy remains an interesting direction for future work.
>
> Finally, we also examined the effect of adjusting the regularization coefficients ($\alpha$, $\beta$). We observed that these coefficients primarily affect the early stage of training, and that using fixed values tends to hinder convergence or introduce instability. As in Appendix D, we found that simple monotone schedules such as logistic or linear scheduling provide more stable training than cyclic or cosine annealing. Although the optimal choice is data-dependent, our experiments indicate that monotone schedules serve as a reliable and practical default.

---

### Meta-Review · Area_Chair_Lfhi · 2026-01-03

**Summary:**

To incorporate robustness in generative modeling this paper introduces a new class of multivariate power-law distributions which can be viewed as an norm-based counterpart of the multivariate distribution.  By the joint minimization view of variational inference, the authors propose the ParetoVAE, a probabilistic autoencoder that minimizes the gamma-power divergence between two statistical manifolds. ParetoVAE employs the symPareto distribution for both prior and encoder, with flexible decoder options including Student's and symPareto distributions. The experiments show ParetoVAE's effectiveness across multiple domains. The reviewers indicate that this paper has a strong conceptual motivation and that the proposed approach is novel. Furthermore, the method is theoretically grounded and the empirically validated across several benchmarks. They also indicate that the paper is clearly written. However, in spite of this, some reviewers have identified some weaknesses such as a more detailed cost vs. ELBO comparison is missing (partially addressed in the rebuttal), and the lack of ablations on gamma-power choices and sensitivity analysis on hyperparameters (partially addressed in the rebuttal).  The reviewers have also pointed out that the introduction could be improved (the authors facilitated a revision plan) and that there may be a lack of experimental results on real data (the authors indicate that the reason is the computational resources). In spite of this, overall, I believe that this could be an interesting paper for the community.

**Reviewer Concerns:**

The rebuttal addressed partially some concerns about the paper, including computational trade-offs by providing runtime comparisons. The authors also showed hyperparameter stability through sensitivity analysis, and clarified theoretical justifications. The authors in the rebuttal also planned revisions to strengthen the paper. However, some issues remain unresolved, real-data experiments are still limited and concrete examples in the introduction are pending but planned.

**Reviewer Scores:**

I do not think that the reviewers would have significantly changed their scores.

---

### Decision · Program_Chairs · 2026-01-26

Accept (Poster)